# Concurrent 3D super resolution on intensity and segmentation maps improves detection of structural effects in neurodegenerative disease

## Abstract

We propose a new perceptual super resolution (PSR) method for 3D neuroimaging and evaluate its performance in detecting brain changes due to neurodegenerative disease. The method, concurrent super resolution and segmentation (CSRS), is trained on volumetric brain data to consistently upsample both an image intensity channel and associated segmentation labels. The simultaneous nature of the method improves not only the resolution of the images but also the resolution of associated segmentations thereby making the approach directly applicable to existing labeled datasets. One challenge to real world evaluation of SR methods such as CSRS is the lack of high resolution ground truth in the target application data: clinical neuroimages. We therefore evaluate CSRS effectiveness in an adjacent, clinically relevant signal detection problem: quantifying cross-sectional and longitudinal change across a set of phenotypically heterogeneous but related disorders that exhibit known and differentiable patterns of brain atrophy. We contrast several 3D PSR loss functions in this paradigm and show that CSRS consistently increases the ability to detect regional atrophy both longitudinally and cross-sectionally in each of five related diseases.

## 1 Introduction

Magnetic resonance image (MRI) datasets capturing in vivo longitudinal change in the human brain are currently available at unprecedented scale. These data allow us to quantify the complex etiology of neurodegenerative disease during life. A fundamental problem in quantifying brain disorders from imaging is that many anatomical structures are small in comparison to image resolution. This is caused by not only limited image resolution but also the potentially convoluted shape of the targeted anatomy [1]. Thinner, more oblate and/or curved structures undergo more distortion due to sampling-related aliasing in comparison to larger, more spherical structures. These distortions can limit detection power in the context of either clinical trials and/or at the level of patient specific medicine [2, 3]. These results also show that, based on first principles, many disease relevant anatomical structures in the brain, in particular cortical regions, mid-brain regions and hippocampal subfields, should be quantified at higher resolutions (e.g. $\approx 0.5\text{mm}^3$ or smaller rather than the more commonly available $\approx 1\text{mm}^3$). The need for increased resolution is only heightened when considering aging and neurodegeneration where some brain structures may lose half or more of their pre-disease onset volume or thickness.

Perceptual super resolution (PSR) for 2D RGB imagery consistently demonstrates the ability to estimate more "realistic" looking upsampled data in comparison to traditional linear or nearest neighbor interpolants [4]. While many competitive methods are available, the deep back projection network (DBPN) [5] performed consistently in several competitions including NTIRE 2018 and 2019

[6], AIM 2019 [7] and PIRM 2018 [8]). These large challenges compared dozens of methods with respect to a variety of both perceptual and reconstruction metrics at different levels of upsampling and noise.

Can the 2D RGB performance advantages of methods like the DBPN translate to improvements in the 3D quantification of brain regions as seen in MRI? If so, then PSR for 3D neuroimaging promises to improve quantification by better resolving the brain's internal structures and tissue boundaries. While traditional evaluations of PSR focus on reconstruction error and perceptual impression, these measurements do not provide clinically relevant evidence of PSR's value in quantification. One barrier to evaluating PSR's impact on clinically relevant outcomes (segmentation volumes) is that ground truth segmentations do not exist at the super-resolved scale. To address this concern, [9] simulated low resolution magnetic resonance images (MRI) of the brain from high-resolution (HR) images obtained from the Human Connectome Project [10, 11]. They then applied a very deep super resolution (VDSR) model to the simulated data and the high-resolution data and compared the accuracy of an automated cortical segmentation method. This careful evaluation study demonstrated that cortical segmentation on the VDSR images closely approximated the HR data. However, relatively few details are provided about the training of this model and associated loss functions. Furthermore, it remains unclear whether these improvements in reconstruction error would translate to the detection of population-level effects in real world data particularly in the aging populations that are the target of the majority of interventional trials for the brain.

A more recent effort in volumetric PSR for medical images [12] proposed SOUP-GAN: Super-resolution Optimized Using Perceptual-tuned Generative Adversarial Network (GAN). SOUP-GAN adopts transfer learning from 2D VGG19 to 3D as proposed in [13] to produce a pseudo-volumetric perceptual metric [14]. Shan et al. used this metric to denoise low-dose computed tomography (CT) images and showed its effectiveness at preserving small anatomical structures. Similarly, the SOUP-GAN effort demonstrates that the pseudo-3D perceptual metric improves both PSNR and SSIM as well as shows visually appealing upsampling for a variety of medical imaging modalities. That is, the surprising utility (in 2D) of VGG weights as a feature space [15] appears to at least partially transfer to PSR in 3D medical imaging.

The current research provides perhaps the first broadly scoped, real world evaluation of MRI PSR for quantification of neurodegenerative disease. Moreover, we demonstrate that a regression network (ResNet) that predicts T1w image quality can yield a directly useful perceptual feature space that performs competitively with pseudo-3D VGG19 features. We build these contributions upon the backbone of a set of methods that we call concurrent super resolution and segmentation (CSRS) that extends the proven 2D DBPN to 3D and also includes extra output channel(s) enabling segmentation maps to be upsampled concurrently. We use this framework to test the impact of different loss functions on a set of domain-specific, clinically relevant segmentation measurements related to brain atrophy. Specifically, we evaluate CSRS on the quantification of frontotemporal disorders [3] from publicly available longitudinal T1-weighted (T1w) neuroimaging (i.e. MRI). Of the several combinations of losses that we evaluate, the best model improves not only segmentation performance (when ground truth is available) but also detection power across all our related disorders: structural changes in behavioral variant frontotemporal dementia (bvFTD), semantic variant primary progressive aphasia (svPPA), nonfluent/agrammatic PPA (naPPA), progressive supranuclear palsy (PSP) and corticobasal syndrome (CBS) each of which impacts known networks in the brain. CSRS with a new perceptual loss based on a shallow ResNet layer performs as well or better than VGG-based models in this test of the practical usefulness of PSR.

The primary contributions of this work include:

- new PSR that upsamples multi-label segmentations at the same time as intensity;
- a new real world evaluation paradigm for PSR in neuroimaging;
- comparison of three perceptual loss functions for PSR, two of which are new;
- demonstration that loss choice impacts detection power in natural history studies of neurodegenerative disease. Standard intensity similarity and segmentation overlap metrics, on the other hand, do not discriminate performance between the candidate CSRS options.

Model weights, sample data, and training code will be made publicly available after anonymous review.

# 2  Methods

**Software platform**: We employ the `ANTsX` platform [16] version 2.3.5 for anatomical labeling, data augmentation/sampling during model training and to form the tabular data for the statistical evaluation. All MRI processing details follow [16]. `Tensorflow` 2.6.2 is used for deep learning including a ResNet implementation and the CSRS architecture. `R` version 4.1 is used for statistical analysis with packages `lmer` and `ggplot2`. All MRI processing was done on Amazon Web Services `parallel cluster` with 24 cores and 32GB RAM per process (Intel(R) Xeon(R) Platinum 8259CL CPU @ 2.50GHz).

**Data: Human Connectome Project (HCP):** We downloaded 1,113 high-resolution $0.7mm^3$ T1-weighted images from the HCP on which to train CSRS. These T1w data were acquired using a magnetization-prepared rapid gradient-echo (MPRAGE) sequence on a customized 3T Siemens Skyra; see [10] for all details of acquisition. As such, these images provide both high resolution and high quality in comparison to the majority of publicly available T1w MRI. Critically, they provide superior resolution for the thin convoluted cortical layer that is critical to the measurement of brain atrophy in frontotemporal disorders. We transformed these data into `numpy` blocks with randomly selected high-resolution $64^3$ patches and paired low-resolution $32^3$ patches. For each patch pair, we also provide a high-resolution binary segmentation and a low-resolution downsampled version of that binary segmentation. Each patch segmentation was gained by 2-class k-means performed on the patch where the center voxel's label determines which class (1 or 2) is used as foreground. This collection of 16,640 patches is then divided randomly into train (n=16,384) and test sets.

**Data: Parkinson's Progression Markers Initiative (PPMI):** PPMI is a longitudinal multi-center clinical study of PD patients and age-matched healthy controls `http://www.ppmi-info.org`. PPMI employed(s) over 20 data collection sites with scanners that span the primary manufacturers (Siemens, GE, Phillips), a variety of head coils and also magnet strengths (1.5T, 3T). This heterogeneity of data collection provides a rich set of T1w images with highly variable image contrast, resolution and quality. We manually reviewed and labelled 1,431 raw T1w from PPMI to capture the range of quality in an ordinal scale. This resulted in a ground truth dataset with 456 images given grade "A" (superior), 568 given grade "B", 350 given grade "C" and 57 given grade "F" which represents images that are of little to no use for quantitative studies of brain structure. We then employed a standard 3D ResNet (`antspynet.create_resnet_model_3d` with parameters `lowest_resolution=32`, `number_of_classification_labels=4`, `cardinality=1`, 39,424,004 parameters, 53 3D convolutional layers) to learn to predict this scale automatically and reliably from the input T1w. We denote this network as a T1w Quality Rating Resnet (T1wQRRResnet). Details of training T1wQRRResNet are in Supplementary Information.

**Data: Frontotemporal Lobar Degeneration Neuroimaging Initiative (NIFD) & 4-Repeat Tauopathy Neuroimaging Initiative (4RTNI)**: These inter-related multi-site studies share the goal of improving the quantification of frontotemporal spectrum disorders with both imaging and clinical scores. Like PPMI and HCP, these studies provide longitudinal T1w images that enable measurement of not only the baseline brain structure differences between controls (individuals without a disease i.e. normal aging) and disease groups but also differences in rates of change due to neurodegeneration. We downloaded and curated 4RTNI and NIFD T1w data and merged these images into a common database. These images were collected at three different sites using protocols consistent with ADNI 3T guidelines [2]. The images overall have a median spacing that is isotropically 1mm with a minority of subjects with out-of-plane spacing up to 1.2mm. As such, these data suit the goals of testing PSR for benefits to the quantification of neurodegenerative disease. After filtering data for very low quality images and the presence of longitudinal data collected within 2 years of baseline, we obtained 128 baseline/171 followup images for controls, 60/112 for bvFTD, 38/72 for naPPA, 37/71 for svPPA, 55/70 for CBS and 75/102 for PSP. Further cohort details (age, education, sex, etc) are available in supplementary information. We processed all images consistently and automatically with default ANTsX pipelines to gain cortical, medial temporal lobe and deep brain structure segmentations for every subject as described in [16]. By consensus, co-authors selected a priori regions for testing within each of four groups CBS/PSP [17], bvFTD, svPPA and naPPA [18–24]. Details of the regions and rationale for their selection are available in the Supplementary Information. See Figure 1 for an overview of processing, the CSRS method and a visualization of the regions (1.C).

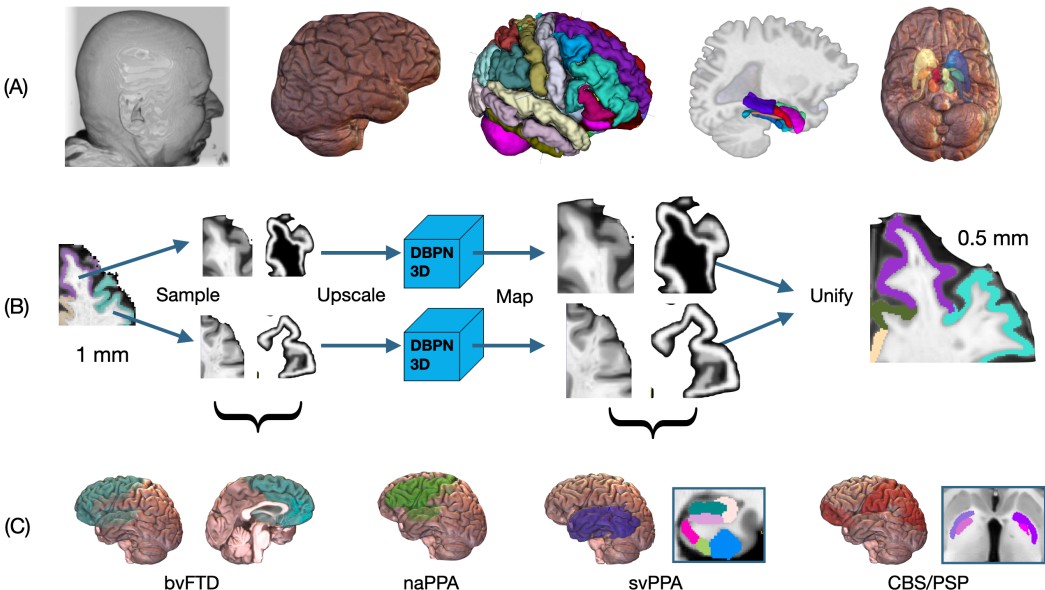

Figure 1: (A) Image processing begins with raw MRI, extracts the brain, labels cortical regions, labels medial temporal lobe regions and labels deep brain regions. (B) The CSRS method is used, here, to upsample data by a factor of 2 isotropically; the sketch of the algorithm provides an example of how two nearby regions would flow through the method and be stitched back together at high resolution. (C) The impact of SR on quantifying neurodegeneration is assessed on a priori regions that are specific to each clinical diagnostic group; all regions are bilateral except for svPPA which uses only left hemisphere cortical and medial temporal labels.

## 2.1 Concurrent super resolution and segmentation methods

CSRS uses, as a sub-algorithm, a three-dimensional and multi-output version of the neural network architecture defined by the 2D deep back projection network (DBPN) [5]. The DBPN is uniquely relevant to medical imaging in that it is perhaps the first published SR method that integrates the downsampling-upsampling error (i.e. residual layers) as a feature map. This novel architecture may prevent feature hallucination and constrain the high-resolution image to maintain features that are consistent with the low-resolution input. We extend the 2D DBPN to 3D MRI data by, first, translating 2D convolutions, padding, striding and other relevant parameters to 3D. To generalize the architecture further, we allow options for not only convolutional upsampling (transposed convolution) but also nearest neighbor (or linear) interpolation layers at the user's choice. Lastly, we implement flexible choices of input channels, the number of residual layers (backprojection) layers and the number of outputs. This 3D DBPN network implementation is available within R and python. All parameters were the same as the published work [25] (though in translation to 3D) with the exception of the number of back projection layers which has a large impact on the number of parameters. We reduced the number of backprojection layers to 5 (16,264,322 parameters) due to the memory limitations caused by working with large 3D images and limited GPU resources (all GPU computations in this work were implemented with Nvidia V100s locally).

Efficient computational strategy is essential for CSRS to be applied to large 3D images (a brain image may contain 10 million voxels) when CPUs and RAM are limited. As such, a local patch-work strategy is necessary. Sampling, upsampling, mapping and unification (SUMU) are the common steps needed for not only training but also inference. "Sampling" decides the form of the input data: full images (not used here), image patches (used here in training) or anatomical image regions (used here in inference). Upscaling determines the core approach to transferring the low-resolution data to a higher-resolution output. Mapping compensates for shape or intensity distortion. Finally, "unification" is an ensembling or merging step that brings together several sub-estimates of an SR image into a single joined (final/full) SR image. We detail each of the 4 components below.

*Sampling*: We choose a patch-based model for training as these can easily be applied to input data with different resolutions and fields of view. A second reason for patch-based modeling is that a candidate network does not need to learn the full scope of image variation. This results in shallower and faster to train networks that fit more easily onto readily available GPUs. The choices made during sampling step define the feature basis set. Because prior super-resolution competitions suggest larger patches lead to better performance, we choose the largest patches that would permit efficient batch sizes of 4 (64x64x64). An additional ad hoc support for this choice is that cortical features are relatively well-resolved in sub-1mm training images when voxel cubes of this sized are used. However, there is no direct evidence that this size of patch domain is optimal for this problem.

*Upscaling*: is done with the CSRS's DBPN architecture using nearest neighbor interpolation for the upsampling layers. The software interface to CSRS also allows the user to optionally employ standard linear (tri-linear) interpolation. We use the linear option as a reference in evaluation studies below.

*Mapping*: may be used to compensate for distortions in the image shape or intensity space. Because each patch is scaled independently on training data (to have an intensity range of -127.5 to 127.5), the output of the PSR upsampled image intensities must be mapped back to the original quantitative space. This is performed by directly comparing the output of the PSR upsampled patch/region to the original data upsampled by nearest neighbor or linear interpolation. As such, we can accurately retain quantitative intensity data at the original scale/units with minimal distortion and/or stitching artifacts.

*Unification*: this is a general term that, here, refers to the algorithm that is used to derive a single CSRS image and multi-label segmentation from multiple CSRS sub-images (not necessarily isotropic patches as in training). In 2D, multiple input images are typically generated from a single input by "augmentation" e.g. random flipping, translation, etc thus allowing a practitioner to gain multiple "votes" about how the SR image should appear at any given voxel. Such a step is used in most PSR competitions to reduce aliasing or artifacts and may involve averaging, sharpening or more complex modeling such as joint intensity fusion, multi-channel deep learning or other ensemble methods. Due to the high memory and computation cost of running CSRS on 3D images, we instead apply CSRS to either sub-regions of interest or, when a full T1w brain image is desired, each hemisphere. The unification step then maps each local patch intensity range back to the original MRI range and then joins the sub-regions back together to complete the SR reconstruction. Augmentation can be employed beyond this but at substantial increase in computation time (e.g. 10x to see meaningful gains due to augmentation).

*Loss functions for CSRS*: We employ a loss function that seeks to balance reconstruction error (intensity difference, abbreviated here as R), edge preserving denoising (total variation, abbreviated as TV), perceptual quality (based on VGG or ResNet) and segmentation overlap (Dice, abbreviated as D). Each of these terms can be up or down weighted to control the network's performance where mean squared error (L2 intensity error) leads to smoother results, L1 (or total variation) provides denoising and the perceptual loss yields more natural appearing output textures and shapes. The Dice loss term seeks to minimize distortions in the shape of segmentation objects on the output of CSRS. The Dice loss is only applied to the second output channel of the network which uses a sigmoid activation function appropriate for probabilistic/binary data. We refer to CSRS trained with specific combinations of these losses by concatenation of the abbreviations above. For example, CSRS.R.TV.D.Res6 refers to a network trained with reconstruction loss, TV regularization, Dice loss and the 6th layer of the T1wQRResNet for perceptual loss.

Recent research demonstrates that deep learning models trained on large-scale object detection reference datasets (e.g. `imagenet`) encode a feature space that may mimic human perception [26]. Such perceptual spaces typically arise from the activations that occur within the layers of convolutional networks trained on massive classification datasets. Here, however, we compare a standard VGG based perceptual space (`block2_conv2`) (mapped to 3D as described before) to those defined by the T1wQRResNet. From T1wQRResNet, we choose two different deep layers that have similar numbers of parameters to the 3D version of the VGG19 `block2_conv2` network: `res_conv_block_6` (the 2nd convolutional block) and `res_conv_block_21` (the 7th convolutional block). This allows us to compare perceptual metrics based on either pseudo-3D VGG19 or our intrinsically 3D `res_conv_block` choices.

Quantification of medical images requires a high degree of faithfulness to the input data. "Halluci-nated" features are undesirable. As such, our baseline loss function focuses on reconstruction error

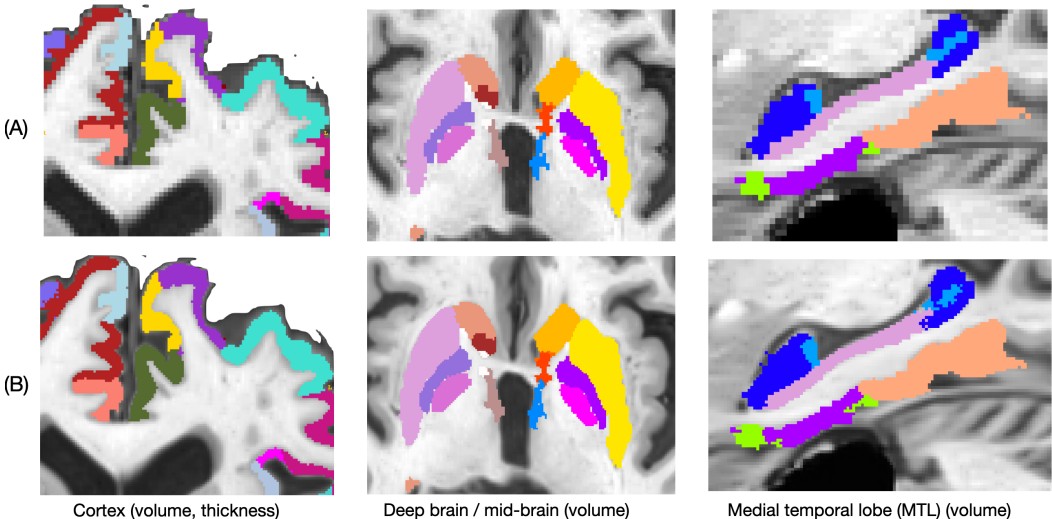

| Cortex (volume, thickness) | Deep brain / mid-brain (volume) | Medial temporal lobe (MTL) (volume) |

Figure 2: Best model CSRS applied to three categories of anatomy where row (A) is the original resolution (OR) and segmentation and row (B) is the output of CSRS.R.TV.D.Res6.

and TV for both intensity and segmentation images. We then add perceptual and Dice losses for further comparison. If we denote $I$ as the estimated super-resolution, $I_s$ as the estimated segmentation from the sigmoid output channel, $J$ as the real high resolution image, $J_s$ as the real high resolution segmentation, then the final loss function that we optimize is:

$$\|I-J\|^2\,w_r^i+\|I_s-J_s\|^2\,w_r^s+TV(I,J)\,w_t^i+TV(I_s,J_s)\,w_t^s+\|f_n(I)-f_n(J)\|^2\,w_f+Dice(I_s,J_s)\,w_d$$

where the term $\|\cdot\|$ indicates the euclidean norm, $TV(\cdot,\cdot)$ indicates the total variation norm (which provides denoising), $w_{r,t,f,d}$ (superscripts for intensity or segmentation) indicates a term-specific scalar weight and $f_n(.)$ indicates a perceptual feature map. The weight terms can be tuned for performance and application area given an objective and quantitative evaluation metric. We initially manually tuned the training of a DBPN model with only the reconstruction metrics ($\|I-J\|^2\,w_r^I+\|I_s-J_s\|^2\,w_r^s$ with $w_r^i=5e-4$ and $w_r^s=1$) using `adam` optimizer and learning rate 5e-5. We then set weights relative to the value of the reconstruction error after convergence such that: the TV loss is roughly 2/3 the reconstruction term (R); the perceptual loss is roughly 3x R; the Dice loss is roughly equivalent to the perceptual loss. This strategy, based on our task-specific goals, enables us to compare models consistently and add/subtract terms without extensive weight optimization.

*Computation and inference*: All models were implemented with `tensorflow`. The computation to double magnification – for a single T1w – takes (generally on a modern computational platform) between 10 and 40 minutes. Results are computed region-wise over the set of segmentation labels where CSRS is run on each cropped label and its associated intensity. When multiple regions are used (as is done here), then results are stitched back together while using a linear mapping back to the original intensity space and a `arg_max` operation to define the hard segmentation labels at every voxel in the stitched, joint intensity/probability double magnification space. See Figure 2 for an example result of CSRS as applied to the variety of brain regions in this study. Figure 3 shows a zoomed visual comparison of the impact on intensity and the lack of stitching artifacts.

## 2.2 Quantification of CSRS impact on segmentation and intensity in ground truth data

Evaluation of PSR results on simulated downsampled-upsampled data does not constitute real world conditions. However, for reference, we include evaluation results based on an independent set of labeled brain images [27]. For these images, we downsample with nearest neighbor interpolation and upsample with linear interpolation (for the intensity) and a "generic label" interpolation that is designed for multi-label images [28] thereby allowing us to report standard metrics of Dice overlap, PSNR and SSIM to complement our study of brain atrophy detection. Figure 4 demonstrates example results illustrating this component of our evaluation.

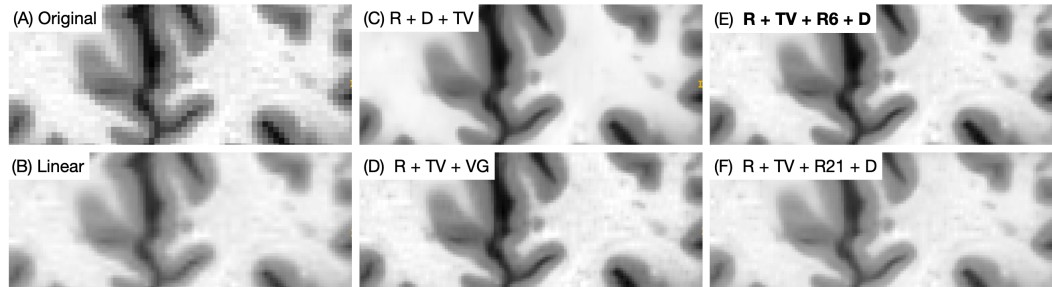

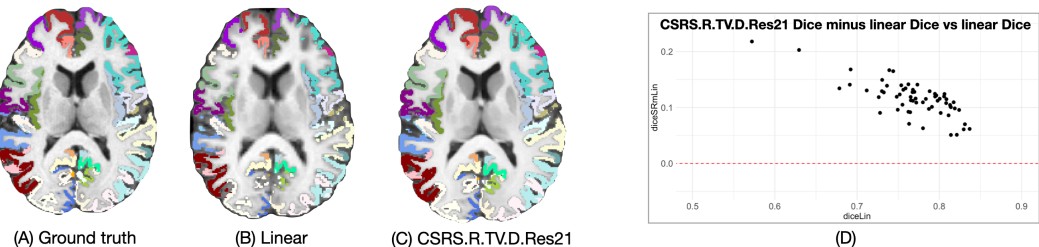

Figure 3: Comparison of CSRS with different loss functions to original resolution and linear upsampling. The bold (panel E) is the best performing model according to quantitative criteria. However, visual differences between the perceptual models (D,E,F) are not easy to discern.

Figure 4: Panel (A) shows the original 1mm³ resolution ground truth image and its segmentation. Panel (B) shows the impact of linear/generic label upsampling of ground truth data artifically downsampled to 2mm³. Panel (C) shows a CSRS result where other models are visually similar to this. Panel (D) demonstrates that all regions improve with CSRS (all differences > 0) and that regions with lower Dice overlap under the linear/generic label model improve more when upsampled with CSRS.

### 2.3 Quantification of effect sizes in frontotemporal disorder atrophy

The frontotemporal disorders produce a profound and debilitating effect on patients with concomitant, symptom-related atrophy. Measuring this atrophy is critical to detecting the effects, for instance, of disease modifying therapies that may slow atrophy. Such measurements are challenged by low resolution and this challenge is compounded by the degeneration process itself.

We use statistical modeling to determine if CSRS can mitigate the known limitations of resolution on atrophy measurement. We adopt an interpretable mixed effects modeling approach (`lmer`)[29] to estimate effect sizes per brain region, per diagnostic category and per resolution/CSRS model. The baseline performance is determined by the effect sizes estimated on the original resolution (OR) data. We estimate effect sizes following [30, 31]. Better methods, under this design, should more reliably identify disease-related atrophy which will be reflected in increased effect sizes for a given set of *a priori* diagnosis-specific regions. The model for the region of interest $i$ ($ROI_i$) is:

$$ROI_i \approx Age_b + Sex + BV_b + DX + \Delta T * DX + (1|ID),$$

with $(1|ID)$ representing a subject-specific random effect, $Age_b$ is the subject's age at the first visit, $BV_b$ is the first visit brain volume, $DX$ is the diagnosis for the subject, $\Delta T$ is the change in time since baseline and the $\Delta T * DX$ represents an interaction between time and diagnosis. The $ROI_i$ represents the volume for all regions. However, for cortical regions, we also use the region's thickness measurement as a second outcome (as this is a standard measurement in morphometry of the human cortex). We estimate effect sizes for cross-sectional effects via the model's parameter fit for the diagnosis ($DX$) term; we estimate longitudinal effect sizes via the parameter on the interaction term.

Table 1: Summary of results where the comparison of the model impact on effect size is computed by bootstrapped (n=1000) paired t.test. The number of pairs is 274 (see Table 2 for further breakdown by category). CSRS losses are abbreviated as R=reconstruction, TV=total variation, D=dice, VGG=VGG19 pseudo 3D features, Res6 is from the 6th layer of T1wQRResNet and Res21 is the 21st layer of T1wQRResNet. srmeanES indicates the mean effect size for the model averaged over all a priori regions; boot.95ci is the 95 percent confidence interval for the improvement in effect size due to the model. t represents the $t$-statistic and boot.p represents the bootstrapped p-value for the significance of the improvement in effect size. Columns psnr and ssim show the standard PSNR and SSIM values for an image for which we have ground truth high-resolution intensity and segmentation. The dice columns show the mean and standard deviation of the Dice overlap between ground truth and the upsampled simulated data with each model, estimated over all regions. Best = **bold**.

| Model | srmeanES | boot.95ci | t | boot.p | psnr | ssim | dice.mean | dice.sd |
|---|---|---|---|---|---|---|---|---|
| OR | 0.559 | 0 / 0 | NA | NA | NA | NA | NA | NA |
| Linear | 0.468 | -0.1006 / -0.08163 | -18.61 | 0 | 40.6 | 0.996 | 0.769 | 0.047 |
| CSRS.R.TV | 0.574 | 0.01124 / 0.01854 | 7.96 | 0 | 42.0 | 0.997 | 0.884 | 0.031 |
| CSRS.R.TV.D | 0.582 | 0.01868 / 0.0273 | 10.38 | 0 | 41.8 | 0.997 | 0.883 | 0.031 |
| CSRS.R.TV.VGG | 0.581 | 0.01775 / 0.02559 | 10.76 | 0 | 42.0 | 0.997 | 0.885 | 0.031 |
| CSRS.R.TV.D.VGG | 0.577 | 0.01403 / 0.02209 | 8.86 | 0 | 41.9 | 0.997 | 0.884 | 0.031 |
| CSRS.R.TV.Res6 | 0.572 | 0.009741 / 0.01665 | 7.49 | 0 | **42.4** | 0.997 | 0.886 | 0.031 |
| CSRS.R.TV.D.Res6 | **0.588** | **0.02497 / 0.0331** | **14.02** | 0 | **42.4** | 0.997 | 0.885 | 0.032 |
| CSRS.R.TV.Res21 | 0.577 | 0.01462 / 0.02143 | 10.40 | 0 | 42.3 | 0.997 | 0.884 | 0.031 |
| CSRS.R.TV.D.Res21 | 0.581 | 0.01814 / 0.02627 | 10.59 | 0 | 42.3 | 0.997 | **0.887** | 0.03 |

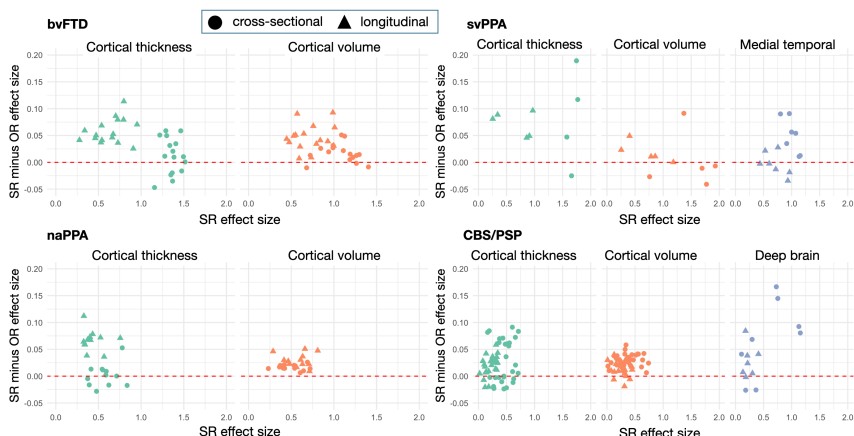

Figure 5: Bland-Altman plots for model CSRS.R.TV.D.Res6 demonstrate variability in the performance by type of anatomy and by diagnostic grouping with some individual points generating substantially greater % improvement than suggested by the overall trend. Similarly, a few points show decreased performance relative to OR.

## 3  Results

Table 1 summarizes overall results where we show original resolution and results from linear upsampling, as baseline, and compare to eight variants of CSRS. Two of these do not use perceptual metrics. The remaining six add or subtract Dice loss and each of our candidate perceptual losses. Table 1 shows both the aggregate impact of model on effect size estimates in the neurodegeneration data as well as intensity similarity (reconstruction) and Dice overlap in the ground truth data. Dice overlap (a measure that varies between zero and one) improves by a margin of 0.11 to 0.123 (95% CI bootstrapped percentile confidence interval, $p < 1e - 16$. See Figure 4.

Table 2 focuses on the two perceptual models with the greatest improvement from original resolution as assessed by pairwise $t$-test. It breaks down the effect size results in relation to which type of effect size is being analyzed (cross-sectional or longitudinal) and by brain region / diagnostic grouping. Relatedly, Figure 5 shows a Bland-Altman style plot that demonstrates, for the CSRS.R.TV.D.Res6 model, the range of effect size changes due to CSRS across all 274 measurement points.

Table 2: Summary of results for the two best perceptual models broken down by anatomical class, type of predictor (longitudinal or cross-sectional) and diagnostic groups. The n column indicates the number of samples used in the statistical testing. The codes in the AnatClass column are: CtxV - cortical volume; CtxT - cortical thickness; MB - deep brain (for CBS/PSP); MTL - medial temporal lobe (for svPPA). The columns that have non-NA DX2 means that both DX and DX2 groups were aggregated in the computation of the bootstrapped paired $t$-test for the given group of anatomy.

| Model | AnatClass | isLong | DX | DX2 | n | srmeanES | boot.95ci | t | boot.p |
|---|---|---|---|---|---|---|---|---|---|
| CSRS.R.TV.VGG | All | Both | NA | NA | 274 | 0.581 | 0.01775 / 0.02559 | 10.763 | 0.0000 |
| CSRS.R.TV.VGG | CtxV | Cross | bvFTD | naPPA | 28 | 0.826 | 0.007153 / 0.01626 | 4.936 | 0.0000 |
| CSRS.R.TV.VGG | CtxT | Cross | bvFTD | naPPA | 28 | 1.016 | -0.009547 / 0.01006 | 0.033 | 0.9769 |
| CSRS.R.TV.VGG | MB | Cross | CBS/PSP | NA | 8 | 0.600 | 0.03683 / 0.1208 | 3.458 | 0.0246 |
| CSRS.R.TV.VGG | MTL | Cross | svPPA | NA | 7 | 1.003 | 0.02801 / 0.06981 | 4.190 | 0.0112 |
| CSRS.R.TV.VGG | CtxV | Long | bvFTD | naPPA | 28 | 0.645 | 0.01848 / 0.03313 | 6.719 | 0.0000 |
| CSRS.R.TV.VGG | CtxT | Long | bvFTD | naPPA | 28 | 0.527 | 0.03521 / 0.05297 | 9.445 | 0.0000 |
| CSRS.R.TV.VGG | MB | Long | CBS/PSP | NA | 8 | 0.201 | 0.01409 / 0.03941 | 3.924 | 0.0110 |
| CSRS.R.TV.VGG | MTL | Long | svPPA | NA | 7 | 0.701 | -0.0159 / 0.002513 | -1.309 | 0.2652 |
| CSRS.R.TV.D.Res6 | All | Both | All | NA | 274 | 0.588 | 0.02497 / 0.0331 | 14.021 | 0.0000 |
| CSRS.R.TV.D.Res6 | CtxV | Cross | bvFTD | naPPA | 28 | 0.831 | 0.01199 / 0.02182 | 6.573 | 0.0000 |
| CSRS.R.TV.D.Res6 | CtxT | Cross | bvFTD | naPPA | 28 | 1.023 | -0.002896 / 0.01832 | 1.394 | 0.1604 |
| CSRS.R.TV.D.Res6 | MB | Cross | CBS/PSP | NA | 8 | 0.589 | 0.02166 / 0.1131 | 2.718 | 0.0454 |
| CSRS.R.TV.D.Res6 | MTL | Cross | svPPA | NA | 7 | 1.004 | 0.02782 / 0.07253 | 4.021 | 0.0102 |
| CSRS.R.TV.D.Res6 | CtxV | Long | bvFTD | naPPA | 28 | 0.658 | 0.03203 / 0.04756 | 9.751 | 0.0000 |
| CSRS.R.TV.D.Res6 | CtxT | Long | bvFTD | naPPA | 28 | 0.545 | 0.05421 / 0.06995 | 15.151 | 0.0000 |
| CSRS.R.TV.D.Res6 | MB | Long | CBS/PSP | NA | 8 | 0.198 | 0.006017 / 0.04464 | 2.233 | 0.0166 |
| CSRS.R.TV.D.Res6 | MTL | Long | svPPA | NA | 7 | 0.704 | -0.0177 / 0.01214 | -0.369 | 0.7511 |

## 4  Discussion

The PSNR and SSIM improve similarly across all CSRS models and do not substantively differentiate performance. Dice overlap is consistently superior than linear upsampling across all models but shows little difference between models with perhaps a small advantage for the ResNet features. Greater stratification may be seen when looking at results that relate to quantifying the phenotypic hetero-geneity of brain atrophy in frontotemporal spectrum diagnostic groups. Model CSRS.R.TV.D.Res6 stands out under this criteria with Table 2 suggesting that the majority of the improvement arises for cortical measurements, particularly longitudinally. Performance improvements are not, however, perfectly consistent. Figure 5 shows that CSRS augments effect size in the large majority of regions (some greatly so) but a few regions are subtly better at OR. Additional discussion of performance implications with respect to individual regions and diagnoses is in supplementary information.

The extension of PSR to 3D raises opportunities as well as challenges. Parameter exploration is fundamentally limited because training a model on our patch dataset for 1 epoch takes over 12 hours (we trained each model for 2 epochs or until convergence). Other architectures than DBPN may perform better with CSRS such as ESRGAN [32] or, potentially, methods with stronger modality specific priors on the convolutional kernels [33]. Specifically, fast-training, fewer parameter models may ease some of the computational burden and facilitate more parameter exploration.

CSRS performance is fundamentally limited by the quality of its segmentation inputs. It may be more beneficial to develop new methods that operate at high resolution (HR) – adding substantial computational cost if the goal is to take advantage of HR features – or that take advantage of intrinsically HR ground truth data. The primary barrier to such an effort is the current lack of HR ground truth labels for neuroimaging and in particular for neurodegenerative disease. Moreover, most methods embed resolution assumptions in their own processing choices and optimize for these choices. As such, CSRS bridges a performance gap with a practical solution readily available today.

Retooling existing methods and segmentation labels for HR (e.g. 7T MRI) is costly both computation-ally and in terms of the effort of human experts due to the already high volume of 3D neuroimaging. We demonstrated that CSRS, in most of its variants, leads to significant performance improvements over our reference of original resolution ($1mm^3$) image processing and ground truth labels. Because CSRS operates on existing images and labels, new HR method and segmentation development is not required. Thus, CSRS may be used to improve existing ground truth datasets and existing processed data, today. However, comparison to other and/or larger real world datasets is needed to help determine the extent to which our results may be deployed to new data without concern.

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
