# Concurrent 3D super resolution on intensity and segmentation maps improves detection of structural effects in neurodegenerative disease: Supplementary information

author

26 May, 2022

# 1 Supplementary Information Overview

This document provides supplemental information to the main article detailing:

- the software stack;
- data licenses/acknowledgements;
- pointers to example data and its download location;
- the code used in data generation and training;
- training the quality prediction ResNet;
- the code for evaluating segmentation performance in simulated (downsampled) data;
- cohort details for each diagnostic group;
- the a priori regions selected for each diagnostic group;
- the source data for key results;
- other performance implications with respect to individual regions and diagnoses.

Each point outlined above will be covered in a specific sub-section following the same (shortened) title. The download location for the majority of the supplemental resources, data and tables is at **this link**. The data location may be referenced via DOI:10.6084/M9.FIGSHARE.19848670.V13 [1]. NOTE: this DOI is versioned; e.g. V13 can be updated to V14 when changes are necessary. Please check that you have the latest version.

## 1.1 List of included resources

The resources at the above link include:

- demog.csv - the basic demographic data for 4RTNI and NIFD subjects

- `mindboggle_sr_test.py` - example segmentation evaluation code and super-res code.
  - this script was used to generate data in Table 1 and Figure 4

- OASIS-TRT-20-9_gt_img.nii.gz - the example T1w MRI used in `mindboggle_sr_test.py`

- OASIS-TRT-20-9_gt_seg.nii.gz - the example T1w MRI segmentation used in `mindboggle_sr_test.py`

- neurips_2022_train_SR_fs.R - model training code.

- neurips_2022_data_gen_SR_fs.R - the data generation code.

- CSRS.R.TV.D.Res6.csv - the effect size estimates by diagnosis for the best model (Tables 1 and 2), the linear upsampling (worst) model and the original resolution model;

- CSRS.R.TV.D.h5 - the CSRS model with reconstruction, total variation and Dice losses;

- CSRS.R.TV.D.Res6.h5 - the CSRS model with reconstruction, total variation, Res6 and Dice losses;

- CSRS.R.TV.VGG.h5 - the model with reconstruction, TV and VGG loss.

- eco_sr_supplemental.Rmd - the source code for this document

# 2 Software stack

The software stack uses a mix of `R` (for training and data generation) and `python` for testing and scaled-up application of CSRS and supporting tools. The deep learning core software is `tensorflow`. Most of the `R` code can be implemented in a near direct translation to `python`. The primary tools are available in the `ANTsX` software stack **here** at `github`. Docker containers are also available that package the core tools. Several installation and usage options are available and potential users can contact developers for details if they encounter barriers. The ANTsX software stack operates under the Apache license version 2.0. Key packages used in this work:

- `ANTsR` commit ecfb0b32e340c8b596958a8a0561cf4f310903fa – image processing core tools in `R`;

- `ANTsRNet` commit dcad84e91e9d2f7808e06d9b735b6475717472ea – network architectures in `R`;

- `ANTsPy` pypi release 0.3.2 – image processing core tools in `python`;

- `ANTsPyNet` pypi release 0.1.8 – network architectures in `python`;

- `ANTsPyT1w` pypi release 0.7.1 – image processing standards for volumetric T1-weighted neuroimages in `python`.

These packages have `get_data` commands which supply relevant examples and models for brain image processing.

# 3 Data Acknowledgments

NIFD and 4RTNI and both available via request from the Laboratory of Neuroimaging (LONI). We do not have permission to redistribute this data in raw or derived form but users can download it and process it themselves with `ANTsPyT1w` (part of the `ANTsX` ecosystem) available from `pypi`.

**NIFD/4RTNI:** Data used in preparation of this article were obtained from the Frontotemporal Lobar Degeneration Neuroimaging Initiative (FTLDNI) database http://4rtni-ftldni.ini.usc.edu. The investigators at 4RTNI/NIFD/FTLDNI contributed to the design and implementation of FTLDNI and/or provided data, but did not participate in analysis or writing of this report (unless otherwise listed). Data collection and sharing for this project was funded by the Frontotemporal Lobar Degeneration Neuroimaging Initiative (National Institutes of Health Grant R01 AG032306). The study is coordinated through the University of California, San Francisco, Memory and Aging Center. Frontotemporal Lobar Degeneration Neuroimaging Initiative data are disseminated by the Laboratory for NeuroImaging at the University of Southern California.

Data collection and sharing for this project was funded by the 4-Repeat Tauopathy Neuroimaging Initiative (4RTNI) (National Institutes of Health Grant R01 AG038791) and through generous contributions from the Tau Research Consortium. The study is coordinated through the University of California, San Francisco, Memory and Aging Center. 4RTNI data are disseminated by the Laboratory for Neuro Imaging at the University of Southern California.

Data collection and sharing for this project was funded by the Frontotemporal Lobar Degeneration Neuroimaging Initiative (National Institutes of Health Grant R01 AG032306). The study is coordinated through the University of California, San Francisco, Memory and Aging Center. FTLDNI data are disseminated by the Laboratory for Neuro Imaging at the University of Southern California.

**PPMI:** Data used in the preparation of this article were obtained from the Parkinson's Progression Markers Initiative (PPMI) database www.ppmi-info.org/access-data- specimens/download-data. For up-to-date information on the study, visit ppmi-info.org. PPMI – a public-private partnership – is funded by the Michael J. Fox Foundation for Parkinson's Research and funding partners, including 4D Pharma AbbVie

Inc. AcureX Therapeutics Allergan Amathus Therapeutics Aligning Science Across Parkinson's (ASAP) Avid Radiopharmaceuticals Bial Biotech Biogen BioLegend Bristol Myers Squibb Calico Life Sciences LLC Celgene Corporation DaCapo Brainscience Denali Therapeutics The Edmond J. Safra Foundation Eli Lilly and Company GE Healthcare GlaxoSmithKline Golub Capital Handl Therapeutics Insitro Janssen Pharmaceuticals Lundbeck Merck & Co., Inc. Meso Scale Diagnostics, LLC Neurocrine Biosciences Pfizer Inc. Piramal Imaging Prevail Therapeutics  F. Hoffmann-La Roche Ltd and its affiliated company Genentech Inc. Sanofi Genzyme Servier Takeda Pharmaceutical Company Teva Neuroscience, Inc. UCB Vanqua Bio Verily Life Sciences Voyager Therapeutics, Inc. Yumanity Therapeutics, Inc.

# 4  Reproducing key results

## 4.1  pointers to example data and its download location

The repository DOI:10.6084/M9.FIGSHARE.19848670.V13 contains an example image and segmentation from the **MindBoggle data**. These images can be used with the file `mindboggle_sr_test.py` to implement a basic test of simulated segmentation performance with CSRS. Models `CSRS.R.TV.D.h5`, `CSRS.R.TV.VGG.h5` and `CSRS.R.TV.D.Res6.h5` are available and named as the same way as in the main paper. This same code base is used in the evaluation of 4RTNI and NIFD combined data.

## 4.2  code for data generation and training

In the same location are two files: `neurips_2022_data_gen_SR_fs.R` and `neurips_2022_train_SR_fs.R` As names suggest, these provide the code that was used (first) to generate the full training dataset. Once training data is done, we used the `neurips_2022_train_SR_fs.R` code with the parameters described in the text to train the various versions of our models. Key to this is the `T1wQRResNet` which is called `~/.antspyt1w/resnet_grader.h5` available from the `antspyt1w` data repository.

## 4.3  training the quality prediction ResNet (T1wQRResNet)

A key contribution of the article is a new feature space based on a ResNet that predicts T1-weighted neuroimage quality. The network was trained on brain extracted (`antspyt1w.brain_extraction`) PPMI T1w data to predict the `A`, `B`, `C` and `F` classification labels of quality. The loss function is computed over batches of size 16 with images at $2mm^3$ resolution. Denoting $g_t$ as the true label binary assignment matrix (shape $16 \times 4$), $g_p$ as predicted label `softmax` probability matrix (shape $16 \times 4$) and $g_p^s, g_t^s$ as the numeric coding of those labels (0,1,2,3) (shape $16 \times 1$), then we minimize:

$$m(CCE(g_t, g_p)) - m(pearson(g_t^s, g_p^s)) * \gamma$$

where $m$ indicates a `mean` (or averaging) operation, $\gamma = 0.1$ is a scalar weight, $CCE$ is the categorical cross entropy and `pearson` represents the Pearson correlation. We train the network on augmented PPMI data (random small – 0 to approximately 15 degree across any axis – rotations of the original images) until convergence where convergence was determined on 5% data left out from training with a balanced distribution across grades.

## 4.4  evaluating segmentation performance in simulated (downsampled) data

The file `mindboggle_sr_test.py` implements the following steps:

- read images and normalize the intensity

```
imggt = ants.image_read( "mindboggle_image/OASIS-TRT-20-9_gt_img.nii.gz" )
seggt = ants.image_read( "mindboggle_image/OASIS-TRT-20-9_gt_seg.nii.gz" )
imggt = ants.iMath(imggt,"Normalize")
```

- downsample the images to 2mm by nearest neighbor and upsample again by generic label (the reference for standard upsampled segmentation results)

```
img = imggt.resample_image( [2,2,2], use_voxels=False, interp_type=1 )
seg = ants.resample_image( seggt,  [2,2,2], use_voxels=False, interp_type=1 )
segup = ants.resample_image_to_target( seg, seggt, interp_type='genericLabel')
intup = ants.resample_image_to_target( img, seggt, interp_type='linear')
```

- calculate PSNR, SSIM and overlap for reference results

```
linpsnr = antspynet.psnr(imggt, intup )
linssim = antspynet.ssim(imggt, intup )
lindice = ants.label_overlap_measures( seggt, segup )
lindice.insert(1,'ssim',linssim)
lindice.insert(1,'psnr',linpsnr)
```

- load the tensorflow model and apply to the low-resolution image and labels

```
srmdl = tf.keras.models.load_model( mdlfn )
ssr = antspyt1w.label_and_img_to_sr(
            img,
            seg,
            srmdl,
            return_intensity=True )
```

- calculate PSNR, SSIM and overlap for CSRS results

```
mypsnr = antspynet.psnr( imggt, ssr['super_resolution'] )
myssim = antspynet.ssim( imggt, ssr['super_resolution'] )
myol = ants.label_overlap_measures( seggt, ssr['super_resolution_segmentation'] )
```

This same code is used in the real world evaluation of neurodegenerative disease.

## 4.5   source data for the cohort

The cohort description is within `demog.csv`.

## 4.6   source data for the key result: effect size estimates

The supplementary information focuses on the results contained within the file `CSRS.R.TV.D.Res6.csv` which provides the data associated with the best performing model.

```
library(subtyper)
```

```
## Registered S3 method overwritten by 'parameters':
##   method                         from
##   format.parameters_distribution datawizard
```

```
raweffsz = read.csv( "CSRS.R.TV.D.Res6.csv")
# bvFTD
raweffsz$BVPrior=FALSE
dxsel = grep("frontal", raweffsz$Anat)
dxsel = c( dxsel, grep("nsula", raweffsz$Anat) )
dxsel = c( dxsel, grep("anterior_cingulat", raweffsz$Anat) )
raweffsz$BVPrior[ dxsel ] = TRUE
# PNFA/naPPA
raweffsz$PNFAPrior=FALSE
dxsel = grep( "percul", raweffsz$Anat )
dxsel = c( dxsel, grep("precentral", raweffsz$Anat) )
dxsel = c( dxsel, grep("middle_frontal", raweffsz$Anat ) )
```

```r
dxsel = c( dxsel, grep("triangu", raweffsz$Anat) )
dxsel = c( dxsel, grep("nsula", raweffsz$Anat) )
raweffsz$PNFAPrior[ dxsel ] = TRUE
# SV PPA
raweffsz$SVPrior=FALSE
dxsel = multigrep( c("left","temporal"), raweffsz$Anat, intersect=TRUE )
dxsel = c( dxsel,  multigrep( c("left","mtl"), raweffsz$Anat, intersect=TRUE ) )
dxsel = c( dxsel,  multigrep( c("left","ippoca"), raweffsz$Anat, intersect=TRUE  ) )
raweffsz$SVPrior[ dxsel ] = TRUE
# CBS
raweffsz$CBSPSPPrior = FALSE
dxsel = grep("entral", raweffsz$Anat )
dxsel = c( dxsel, grep("supramarg", raweffsz$Anat ) )
dxsel = c( dxsel, grep("orbitofrontal", raweffsz$Anat ) )
dxsel = c( dxsel, grep("parietal", raweffsz$Anat)  )
dxsel = c( dxsel, grep("_gpi_", raweffsz$Anat)  )
dxsel = c( dxsel, grep("_gpe_", raweffsz$Anat)  )
# dxsel = c( dxsel, intersect( grep("PSP",raweffsz$name), grep("deep_cit168", raweffsz$Anat)  ) )
raweffsz$CBSPSPPrior[ dxsel ] = TRUE

# raweffsz = read.csv( "CSRS.R.TV.VGG.csv")
oreffsz = raweffsz[ raweffsz$Model == "OR", ]
sreffsz = raweffsz[ raweffsz$Model == "SR", ]
```

# 5   Discussion of key results with respect to cohort and anatomy

## 5.1   cohort details for each diagnostic group

| | BV (N=58) | CBS (N=42) | CON (N=127) | PATIENT (OTHER) (N=45) | PNFA (N=37) | PSP (N=60) | SV (N=36) | Overall (N=409) |
|---|---|---|---|---|---|---|---|---|
| **factor(GENDER)** | | | | | | | | |
| 1 | 38 (65.5%) | 23 (54.8%) | 55 (43.3%) | 30 (66.7%) | 17 (45.9%) | 33 (55.0%) | 19 (52.8%) | 215 (52.6%) |
| 2 | 20 (34.5%) | 19 (45.2%) | 72 (56.7%) | 15 (33.3%) | 20 (54.1%) | 27 (45.0%) | 17 (47.2%) | 194 (47.4%) |
| **Age_BL** | | | | | | | | |
| Mean (SD) | 61.6 (6.87) | 66.5 (6.57) | 63.7 (7.40) | 64.8 (7.56) | 68.8 (7.32) | 70.7 (7.58) | 63.6 (6.21) | 65.3 (7.70) |
| Median [Min, Max] | 61.4 [46.1, 75.1] | 67.5 [53.0, 82.0] | 65.0 [37.0, 81.3] | 64.6 [47.6, 85.7] | 69.0 [54.3, 81.4] | 70.5 [55.0, 86.0] | 64.6 [50.5, 73.7] | 65.8 [37.0, 86.0] |
| **EDUCATION** | | | | | | | | |
| Mean (SD) | 18.4 (15.6) | 18.7 (13.7) | 21.9 (18.8) | 17.7 (12.6) | 18.2 (13.9) | 15.7 (4.01) | 18.6 (14.0) | 19.0 (14.9) |
| Median [Min, Max] | 16.0 [8.00, 99.0] | 16.0 [9.00, 99.0] | 18.0 [12.0, 99.0] | 16.0 [12.0, 99.0] | 16.0 [12.0, 99.0] | 16.0 [2.00, 27.0] | 16.5 [12.0, 99.0] | 16.0 [2.00, 99.0] |
| Missing | 0 (0%) | 2 (4.8%) | 0 (0%) | 0 (0%) | 0 (0%) | 2 (3.3%) | 0 (0%) | 4 (1.0%) |
| **deltaTime** | | | | | | | | |
| Mean (SD) | 0 (0) | 0 (0) | 0 (0) | 0 (0) | 0 (0) | 0 (0) | 0 (0) | 0 (0) |
| Median [Min, Max] | 0 [0, 0] | 0 [0, 0] | 0 [0, 0] | 0 [0, 0] | 0 [0, 0] | 0 [0, 0] | 0 [0, 0] | 0 [0, 0] |

Figure 1: NIFD and 4RTNI group characteristics at baseline.

Figure 1 shows the distribution of baseline characteristics across our diagnostic groups. NOTE: PNFA is equivalent to naPPA; SV is equivalent to svPPA and the "Patient(Other)" group was not used. Figure 2 is the longitudinal cohort description.

## 5.2   the a priori regions selected for each diagnostic group

We exclude large regions such as caudate and putamen as we do not expect a priori improvements due to PSR. For cortical measurements, we assess both volume and thickness.

- bvFTD:

| | BV (N=100) | CBS (N=55) | CON (N=267) | PATIENT (OTHER) (N=45) | PNFA (N=78) | PSP (N=79) | SV (N=83) | Overall (N=711) |
|---|---|---|---|---|---|---|---|---|
| **factor(GENDER)** | | | | | | | | |
| 1 | 69 (69.0%) | 32 (58.2%) | 115 (43.1%) | 27 (60.0%) | 37 (47.4%) | 42 (53.2%) | 52 (62.7%) | 374 (52.6%) |
| 2 | 31 (31.0%) | 23 (41.8%) | 152 (56.9%) | 18 (40.0%) | 41 (52.6%) | 37 (46.8%) | 31 (37.3%) | 337 (47.4%) |
| **Age_BL** | | | | | | | | |
| Mean (SD) | 61.7 (6.49) | 66.3 (6.64) | 65.0 (6.74) | 66.3 (9.38) | 68.2 (7.58) | 70.6 (7.50) | 63.4 (6.33) | 65.5 (7.44) |
| Median [Min, Max] | 62.0 [46.2, 75.1] | 67.0 [53.0, 76.0] | 66.0 [37.0, 81.3] | 64.6 [47.6, 85.7] | 67.1 [54.3, 81.4] | 70.0 [56.0, 86.0] | 64.6 [50.5, 73.7] | 66.0 [37.0, 86.0] |
| **EDUCATION** | | | | | | | | |
| Mean (SD) | 19.1 (16.6) | 20.1 (16.2) | 19.6 (13.2) | 16.4 (2.68) | 19.4 (16.2) | 15.8 (4.00) | 18.6 (13.0) | 18.8 (13.2) |
| Median [Min, Max] | 16.0 [8.00, 99.0] | 16.0 [10.0, 99.0] | 18.0 [12.0, 99.0] | 16.0 [12.0, 20.0] | 18.0 [12.0, 99.0] | 16.0 [2.00, 27.0] | 18.0 [10.0, 99.0] | 17.0 [2.00, 99.0] |
| Missing | 0 (0%) | 2 (3.6%) | 0 (0%) | 0 (0%) | 0 (0%) | 0 (0%) | 0 (0%) | 2 (0.3%) |
| **deltaTime** | | | | | | | | |
| Mean (SD) | 1.44 (1.07) | 0.804 (0.344) | 2.44 (1.89) | 1.19 (1.01) | 1.59 (1.13) | 0.726 (0.263) | 1.50 (1.13) | 1.69 (1.51) |
| Median [Min, Max] | 1.07 [0.00549, 6.11] | 0.673 [0.412, 2.25] | 1.89 [0.442, 8.17] | 1.02 [0.0962, 5.41] | 1.18 [0.288, 5.88] | 0.555 [0.448, 1.35] | 1.11 [0.0165, 5.43] | 1.10 [0.00549, 8.17] |

Figure 2: NIFD and 4RTNI group characteristics at all follow up time points.

- cortical: caudal_anterior_cingulatedktcortex, caudal_middle_frontaldktcortex, insuladktcortex, lateral_orbitofrontaldktcortex, medial_orbitofrontaldktcortex, rostral_anterior_cingulatedktcortex rostral_middle_frontaldktcortex, superior_frontaldktcortex
- naPPA:
  - cortical: caudal_middle_frontaldktcortex insuladktcortex pars_opercularisdktcortex pars_triangularisdktcortex" precentraldktcortex rostral_middle_frontaldktcortex"
- svPPA:
  - cortical: inferior_temporaldktcortex middle_temporaldktcortex parahippocampaldktcortex superior_temporaldktcortex transverse_temporaldktcortex
  - medial temporal: alecmtl ca1mtl dg.ca3mtl parahippocampalmtl perirhinalmtl pmecmtl subiculummtl
- CBS:
  - cortical: inferior_parietaldktcortex superior_parietaldktcortex orbitofrontal_cortex
  - deep brain (gp - globus pallidus): vol_bn_gp_gpe_leftcit168 vol_bn_gp_gpe_rightcit168 vol_bn_gp_gpi_leftcit168 vol_bn_gp_gpi_rightcit168
- PSP:
  - cortical: paracentraldktcortex postcentraldktcortex precentraldktcortex

We join the PSP and CBS groups as these are highly related diagnoses. In addition, in this supplemental example, we add supramarginal gyrus as a target region for CBS/PSP.

## 5.3 Sorted comparisons of best regions per diagnosis and per resolution by type of effect (longitudinal or cross-sectional)

The tables below are autogenerated from a function within this compilable document (`eco_sr_supplemental.Rmd`). The function is called `maketables`. Users can modify this to display results in a different way than shown here. these results highlight key regions for each diagnosis and its associated `a priori` hypothesis sets. We focus on the best model `CSRS.R.TV.D.Res6`.

We also display pairwise test results for each diagnosis broken down by longitudinal and cross-sectional groups. Statistical details are shown for each result. Plots are also supplied.

### 5.3.1 Top $n$ effect sizes and pairwise tests for bvFTD

Regions with the highest effect sizes are relatively unsurprising for these bvFTD subjects. However, the "best" regions for cross-sectional detection may not equal the best for longitudinal detection. This is generally true for all groups/diagnoses.

Table 1: OR bvFTD : regions sorted by cross-sectional effect sizes.

|     | Anat | DXBV | deltaTime.DXBV |
| --- | --- | --- | --- |
| 229 | thk_left_caudal_anterior_cingulatedktcortex | 1.514536 | 0.6161465 |
| 301 | thk_left_rostral_anterior_cingulatedktcortex | 1.491225 | 0.6922448 |
| 400 | thk_right_superior_frontaldktcortex | 1.485251 | 0.6457120 |
| 214 | vol_right_superior_frontaldktcortex | 1.412810 | 0.5940318 |

Table 2: CSRS bvFTD : regions sorted by cross-sectional effect sizes.

|     | Anat | DXBV | deltaTime.DXBV |
| --- | --- | --- | --- |
| 231 | thk_left_caudal_anterior_cingulatedktcortex | 1.515278 | 0.7025271 |
| 402 | thk_right_superior_frontaldktcortex | 1.495763 | 0.7252476 |
| 303 | thk_left_rostral_anterior_cingulatedktcortex | 1.475096 | 0.7286681 |
| 261 | thk_left_lateral_orbitofrontaldktcortex | 1.464453 | 0.5296175 |

Table 3: OR bvFTD : regions sorted by longitudinal effect sizes.

|     | Anat | DXBV | deltaTime.DXBV |
| --- | --- | --- | --- |
| 157 | vol_right_insuladktcortex | 1.212387 | 0.9658840 |
| 64 | vol_left_insuladktcortex | 1.282155 | 0.9434986 |
| 136 | vol_right_caudal_anterior_cingulatedktcortex | 0.690373 | 0.8950198 |
| 43 | vol_left_caudal_anterior_cingulatedktcortex | 1.265494 | 0.8892689 |

Table 4: CSRS bvFTD : regions sorted by longitudinal effect sizes.

|     | Anat | DXBV | deltaTime.DXBV |
| --- | --- | --- | --- |
| 66 | vol_left_insuladktcortex | 1.2966768 | 1.0085604 |
| 159 | vol_right_insuladktcortex | 1.2215546 | 0.9969181 |
| 138 | vol_right_caudal_anterior_cingulatedktcortex | 0.6802068 | 0.9878596 |
| 252 | thk_left_insuladktcortex | 1.3484057 | 0.9525358 |

Table 5: BV : cross-sectional CSRS vs OR pairwise t-tests (continued below)

| Test statistic | df | P value | Alternative hypothesis |
| --- | --- | --- | --- |
| 3.133 | 31 | 0.003762 * * | two.sided |

| mean of the differences |
| --- |
| 0.01473 |

Table 7: BV : longitudinal CSRS vs OR pairwise t-tests (continued below)

| Test statistic | df | P value | Alternative hypothesis |
| --- | --- | --- | --- |
| 12.33 | 31 | 1.734e-13 * * * | two.sided |

| mean of the differences |
| --- |
| 0.05237 |

```
## RStudio Community is a great place to get help:
## https://community.rstudio.com/c/tidyverse

## Registered S3 methods overwritten by 'lme4':
##   method                          from
##   cooks.distance.influence.merMod car
##   influence.merMod                car
##   dfbeta.influence.merMod         car
##   dfbetas.influence.merMod        car
```

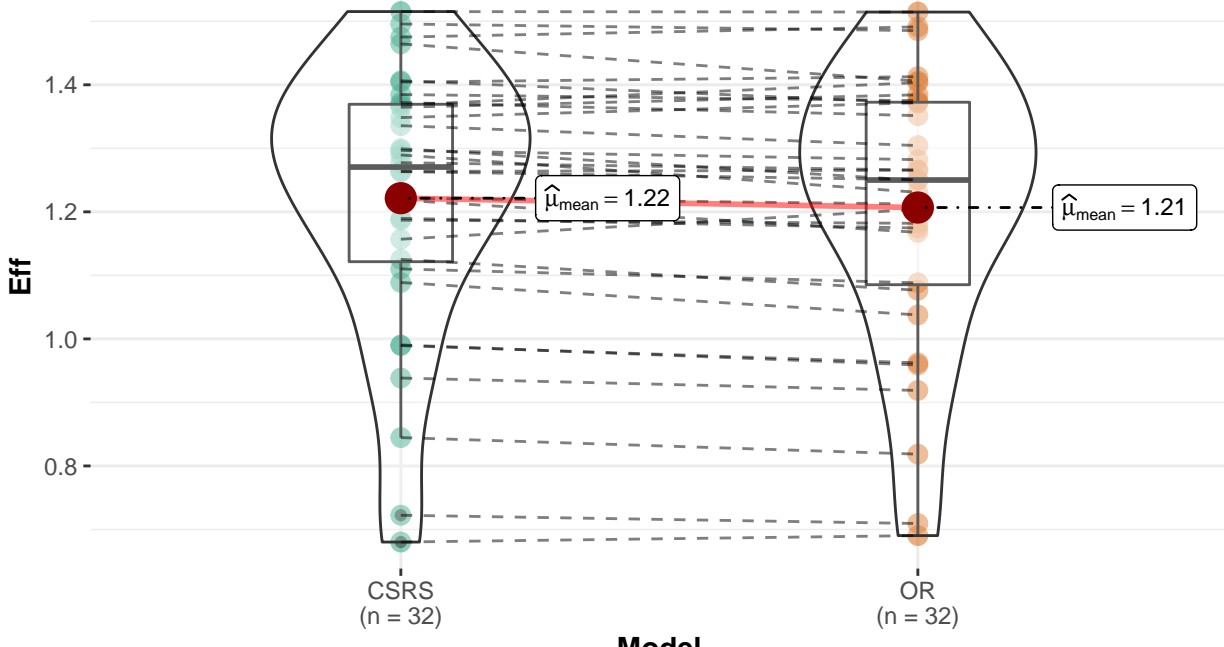

### BV CSRS vs OR: cross−sectional

$t_{\text{Student}}(31) = 3.13, p = 3.76\text{e}{-03}, \widehat{g}_{\text{Hedges}} = 0.54, \text{CI}_{95\%} [0.18, 0.91], n_{\text{pairs}} = 32$

$\widehat{\mu}_{\text{mean}} = 1.22$

$\widehat{\mu}_{\text{mean}} = 1.21$

CSRS (n = 32)   OR (n = 32)

**Model**

**Eff**

$\log_e(\text{BF}_{01}) = -2.32, \widehat{\delta}_{\text{difference}}^{\text{posterior}} = 0.01, \text{CI}_{95\%}^{\text{HDI}} [4.26\text{e}{-03}, 0.02], r_{\text{Cauchy}}^{\text{JZS}} = 0.71$

**BV CSRS vs OR: Longitudinal**

$t_{\text{Student}}(31) = 12.33, p = 1.73\text{e}{-}13, \widehat{g}_{\text{Hedges}} = 2.13, \text{CI}_{95\%} [1.52, 2.79], n_{\text{pairs}} = 32$

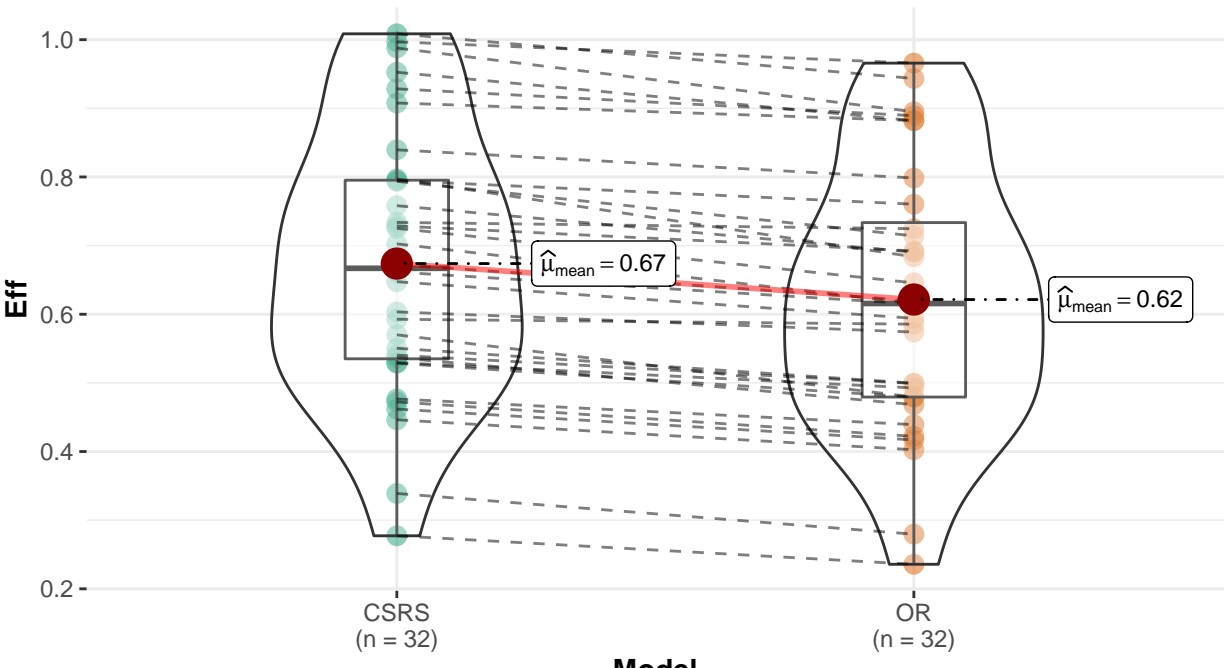

$\log_e(\text{BF}_{01}) = -24.48, \widehat{\delta}_{\text{difference}}^{\text{posterior}} = 0.05, \text{CI}_{95\%}^{\text{HDI}} [0.04, 0.06], r_{\text{Cauchy}}^{\text{JZS}} = 0.71$

### 5.3.2 Top $n$ effect sizes and pairwise tests for naPPA

Note: PNFA = naPPA.

Table 9: OR naPPA : regions sorted by cross-sectional effect sizes.

|     | Anat                                      | DXPNFA    | deltaTime.DXPNFA |
|-----|-------------------------------------------|-----------|------------------|
| 277 | thk_left_pars_opercularisdktcortex        | 0.8532522 | 0.2167136        |
| 232 | thk_left_caudal_middle_frontaldktcortex   | 0.7257261 | 0.3534322        |
| 295 | thk_left_precentraldktcortex              | 0.7133188 | 0.2834232        |
| 109 | vol_left_precentraldktcortex              | 0.7064232 | 0.3855393        |

Table 10: CSRS naPPA : regions sorted by cross-sectional effect sizes.

|     | Anat                                      | DXPNFA    | deltaTime.DXPNFA |
|-----|-------------------------------------------|-----------|------------------|
| 279 | thk_left_pars_opercularisdktcortex        | 0.8360338 | 0.3288237        |
| 234 | thk_left_caudal_middle_frontaldktcortex   | 0.7785160 | 0.4317992        |
| 111 | vol_left_precentraldktcortex              | 0.7204006 | 0.4153661        |
| 297 | thk_left_precentraldktcortex              | 0.7134752 | 0.3422352        |

Table 11: OR naPPA : regions sorted by longitudinal effect sizes.

|     | Anat                                           | DXPNFA    | deltaTime.DXPNFA |
|-----|------------------------------------------------|-----------|------------------|
| 64  | vol_left_insuladktcortex                       | 0.5985578 | 0.7626281        |
| 46  | vol_left_caudal_middle_frontaldktcortex        | 0.6637756 | 0.7040614        |
| 250 | thk_left_insuladktcortex                       | 0.5056475 | 0.6841209        |
| 139 | vol_right_caudal_middle_frontaldktcortex       | 0.3513265 | 0.6840693        |

Table 12: CSRS naPPA : regions sorted by longitudinal effect sizes.

|     | Anat                                           | DXPNFA    | deltaTime.DXPNFA |
|-----|------------------------------------------------|-----------|------------------|
| 66  | vol_left_insuladktcortex                       | 0.6049263 | 0.8101824        |
| 252 | thk_left_insuladktcortex                       | 0.4774403 | 0.7551908        |
| 48  | vol_left_caudal_middle_frontaldktcortex        | 0.6898421 | 0.7132350        |
| 141 | vol_right_caudal_middle_frontaldktcortex       | 0.3712756 | 0.7065073        |

Table 13: PNFA : cross-sectional CSRS vs OR pairwise t-tests (continued below)

| Test statistic | df  | P value    | Alternative hypothesis |
|----------------|-----|------------|------------------------|
| 2.564          | 23  | 0.01736 *  | two.sided              |

| mean of the differences |
|-------------------------|
| 0.00884                 |

Table 15: PNFA : longitudinal CSRS vs OR pairwise t-tests (continued below)

| Test statistic | df | P value | Alternative hypothesis |
|---|---|---|---|
| 9.647 | 23 | 1.501e-09 * * * | two.sided |

| mean of the differences |
|---|
| 0.0484 |

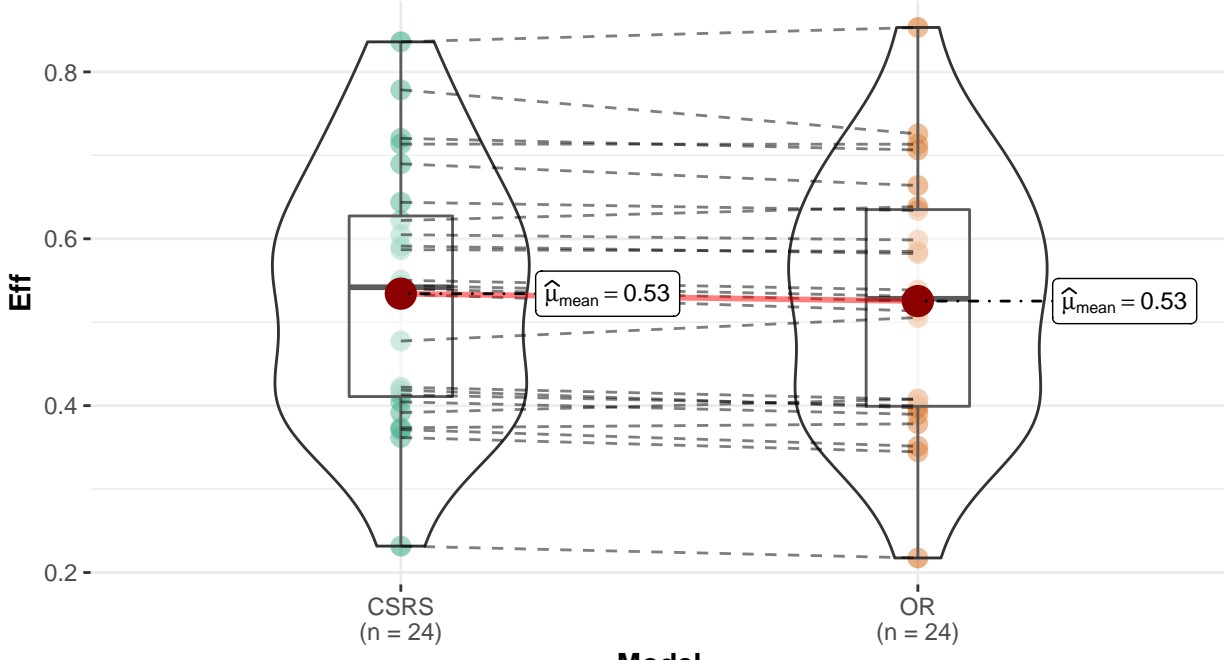

**PNFA CSRS vs OR: cross–sectional**

$t_{\text{Student}}(23) = 2.56$, $p = 0.02$, $\widehat{g}_{\text{Hedges}} = 0.51$, $\text{CI}_{95\%}$ [0.09, 0.93], $n_{\text{pairs}} = 24$

$\log_e(\text{BF}_{01}) = -1.12$, $\widehat{\delta}_{\text{difference}}^{\text{posterior}} = 7.98\text{e}{-}03$, $\text{CI}_{95\%}^{\text{HDI}}$ [1.34e−03, 0.01], $r_{\text{Cauchy}}^{\text{JZS}} = 0.71$

## PNFA CSRS vs OR: Longitudinal

$t_{\text{Student}}(23) = 9.65$, $p = 1.50\text{e}{-}09$, $\widehat{g}_{\text{Hedges}} = 1.90$, $\text{CI}_{95\%}$ [1.25, 2.62], $n_{\text{pairs}} = 24$

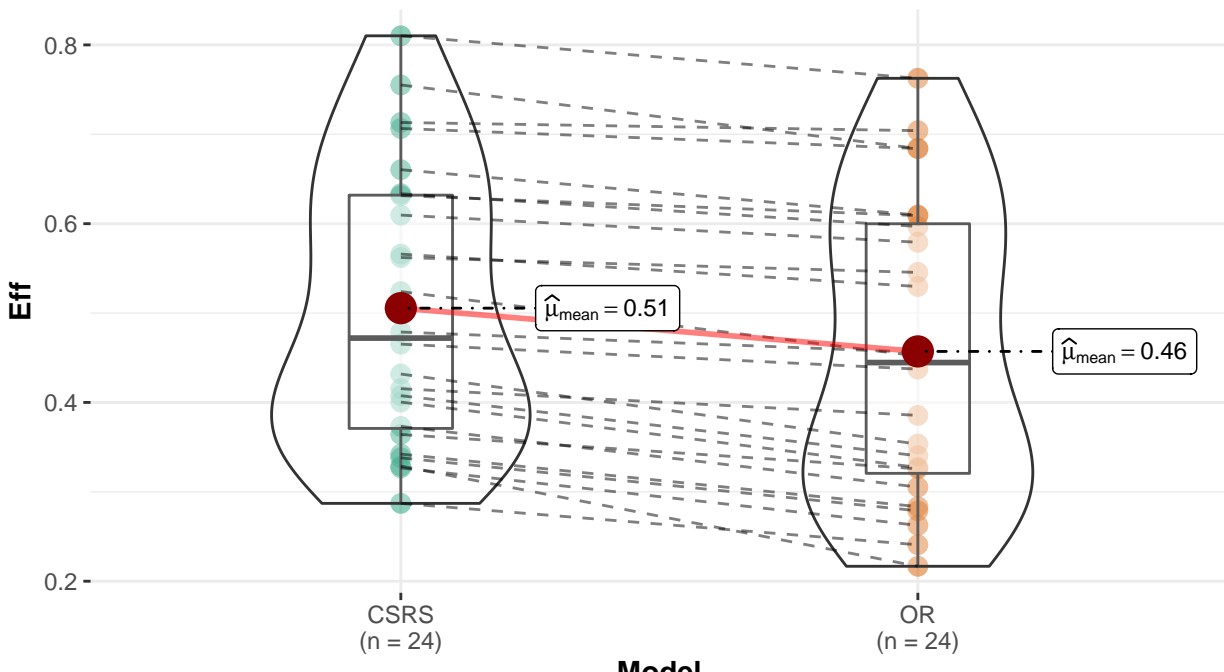

$\log_{e}(\text{BF}_{01}) = -15.84$, $\widehat{\delta}_{\text{difference}}^{\text{posterior}} = 0.05$, $\text{CI}_{95\%}^{\text{HDI}}$ [0.04, 0.06], $r_{\text{Cauchy}}^{\text{JZS}} = 0.71$

### 5.3.3 Top $n$ effect sizes and pairwise tests for svPPA

Note: SV = svPPA.

Table 17: OR svPPA : regions sorted by cross-sectional effect sizes.

|     | Anat | DXSV | deltaTime.DXSV |
|-----|------|------|----------------|
| 61  | vol_left_inferior_temporaldktcortex | 1.936158 | 0.7744236 |
| 127 | vol_left_superior_temporaldktcortex | 1.813257 | 0.8476179 |
| 82  | vol_left_middle_temporaldktcortex | 1.704832 | 1.1813152 |
| 313 | thk_left_superior_temporaldktcortex | 1.683481 | 0.8140000 |

Table 18: CSRS svPPA : regions sorted by cross-sectional effect sizes.

|     | Anat | DXSV | deltaTime.DXSV |
|-----|------|------|----------------|
| 63  | vol_left_inferior_temporaldktcortex | 1.929359 | 0.7851387 |
| 129 | vol_left_superior_temporaldktcortex | 1.772414 | 0.8587171 |
| 249 | thk_left_inferior_temporaldktcortex | 1.768534 | 0.9079394 |
| 276 | thk_left_parahippocampaldktcortex | 1.745901 | 0.3411358 |

Table 19: OR svPPA : regions sorted by longitudinal effect sizes.

|     | Anat | DXSV | deltaTime.DXSV |
|-----|------|------|----------------|
| 82  | vol_left_middle_temporaldktcortex | 1.704832 | 1.1813152 |
| 4   | vol_left_dg.ca3mtl | 1.121621 | 0.9813179 |
| 19  | vol_left_subiculummtl | 1.140113 | 0.9663498 |
| 268 | thk_left_middle_temporaldktcortex | 1.526906 | 0.8717287 |

Table 20: CSRS svPPA : regions sorted by longitudinal effect sizes.

|     | Anat | DXSV | deltaTime.DXSV |
|-----|------|------|----------------|
| 84  | vol_left_middle_temporaldktcortex | 1.693781 | 1.1815049 |
| 270 | thk_left_middle_temporaldktcortex | 1.574064 | 0.9682279 |
| 6   | vol_left_dg.ca3mtl | 1.132567 | 0.9627611 |
| 21  | vol_left_subiculummtl | 1.152631 | 0.9320903 |

Table 21: SV : cross-sectional CSRS vs OR pairwise t-tests (continued below)

| Test statistic | df | P value | Alternative hypothesis |
|:--------------:|:--:|:-------:|:----------------------:|
| 2.315 | 16 | 0.03419 * | two.sided |

| mean of the differences |
|:-----------------------:|
| 0.03667 |

Table 23: SV : longitudinal CSRS vs OR pairwise t-tests (continued below)

| Test statistic | df | P value | Alternative hypothesis |
|---|---|---|---|
| 2.759 | 16 | 0.01398 * | two.sided |

| mean of the differences |
|---|
| 0.02554 |

## SV CSRS vs OR: cross−sectional

$t_{\text{Student}}(16) = 2.32$, $p = 0.03$, $\widehat{g}_{\text{Hedges}} = 0.53$, $\text{CI}_{95\%}$ [0.04, 1.05], $n_{\text{pairs}} = 17$

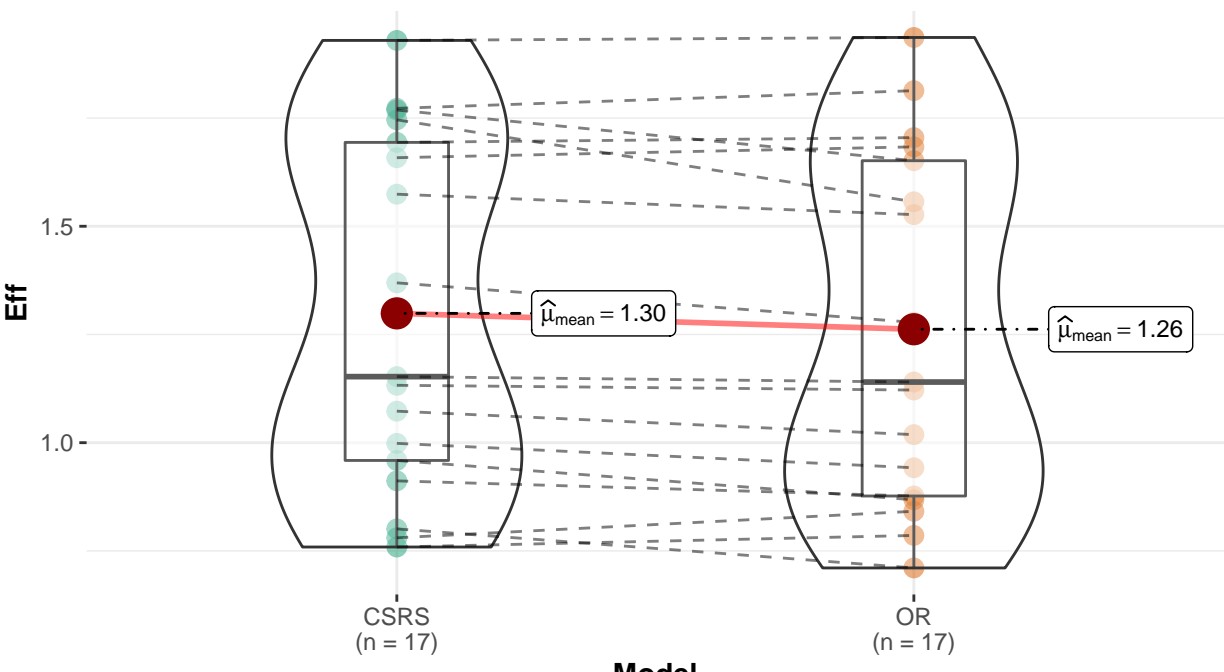

$\log_e(\text{BF}_{01}) = -0.68$, $\widehat{\delta}_{\text{difference}}^{\text{posterior}} = 0.03$, $\text{CI}_{95\%}^{\text{HDI}}$ [2.95e−04, 0.06], $r_{\text{Cauchy}}^{\text{JZS}} = 0.71$

## SV CSRS vs OR: Longitudinal

$t_{\text{Student}}(16) = 2.76, p = 0.01, \widehat{g}_{\text{Hedges}} = 0.64, \text{CI}_{95\%} [0.13, 1.17], n_{\text{pairs}} = 17$

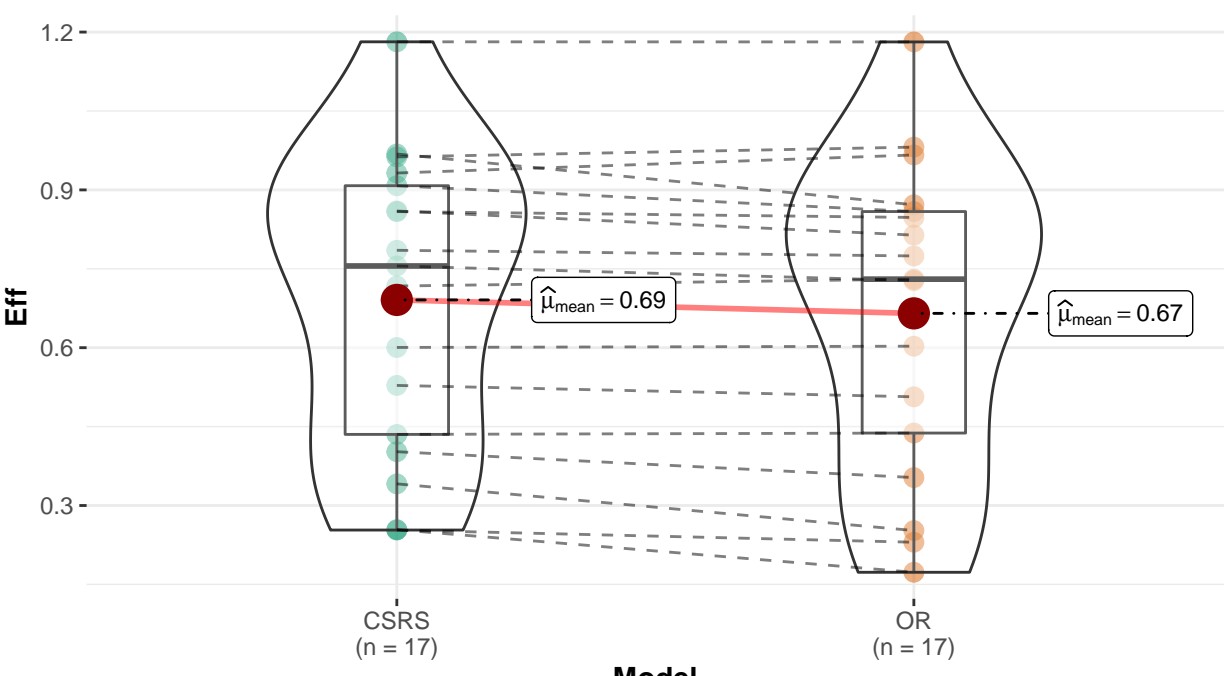

$\log_e(\text{BF}_{01}) = -1.41, \widehat{\delta}_{\text{difference}}^{\text{posterior}} = 0.02, \text{CI}_{95\%}^{\text{HDI}} [4.46\text{e}{-}03, 0.04], r_{\text{Cauchy}}^{\text{JZS}} = 0.71$

### 5.3.4 Top $n$ effect sizes for CBS and pairwise tests

Supramarginal cortex should likely have been included within the original region definitions for CBS and PSP. We therefore include the region here.

Table 25: OR CBS : regions sorted by cross-sectional effect sizes.

|     | Anat | DXCBS | deltaTime.DXCBS |
| --- | --- | --- | --- |
| 316 | thk_left_supramarginaldktcortex | 0.7229556 | 0.3301276 |
| 124 | vol_left_superior_parietaldktcortex | 0.7152843 | 0.4241095 |
| 295 | thk_left_precentraldktcortex | 0.7018938 | 0.2549592 |
| 109 | vol_left_precentraldktcortex | 0.6961477 | 0.3697571 |

Table 26: CSRS CBS : regions sorted by cross-sectional effect sizes.

|     | Anat | DXCBS | deltaTime.DXCBS |
| --- | --- | --- | --- |
| 126 | vol_left_superior_parietaldktcortex | 0.7395759 | 0.4415767 |
| 318 | thk_left_supramarginaldktcortex | 0.7342946 | 0.3803584 |
| 312 | thk_left_superior_parietaldktcortex | 0.7116789 | 0.3430536 |
| 297 | thk_left_precentraldktcortex | 0.7073363 | 0.3134853 |

Table 27: OR CBS : regions sorted by longitudinal effect sizes.

|     | Anat | DXCBS | deltaTime.DXCBS |
| --- | --- | --- | --- |
| 409 | thk_right_supramarginaldktcortex | 0.6269313 | 0.4635650 |
| 130 | vol_left_supramarginaldktcortex | 0.6231034 | 0.4542643 |
| 58  | vol_left_inferior_parietaldktcortex | 0.3984755 | 0.4352320 |
| 124 | vol_left_superior_parietaldktcortex | 0.7152843 | 0.4241095 |

Table 28: CSRS CBS : regions sorted by longitudinal effect sizes.

|     | Anat | DXCBS | deltaTime.DXCBS |
| --- | --- | --- | --- |
| 411 | thk_right_supramarginaldktcortex | 0.6296510 | 0.4999103 |
| 132 | vol_left_supramarginaldktcortex | 0.6283743 | 0.4862934 |
| 60  | vol_left_inferior_parietaldktcortex | 0.4181131 | 0.4472365 |
| 126 | vol_left_superior_parietaldktcortex | 0.7395759 | 0.4415767 |

Table 29: CBS : cross-sectional CSRS vs OR pairwise t-tests (continued below)

| Test statistic | df | P value | Alternative hypothesis |
| --- | --- | --- | --- |
| 4.649 | 35 | 4.611e-05 * * * | two.sided |

| mean of the differences |
| --- |
| 0.0237 |

Table 31: CBS : longitudinal CSRS vs OR pairwise t-tests (continued below)

| Test statistic | df | P value | Alternative hypothesis |
|---|---|---|---|
| 11.44 | 35 | 2.251e-13 * * * | two.sided |

| mean of the differences |
|---|
| 0.02559 |

**CBS CSRS vs OR: cross–sectional**

$t_{\text{Student}}(35) = 4.65, p = 4.61e{-}05, \widehat{g}_{\text{Hedges}} = 0.76, \text{CI}_{95\%} [0.39, 1.14], n_{\text{pairs}} = 36$

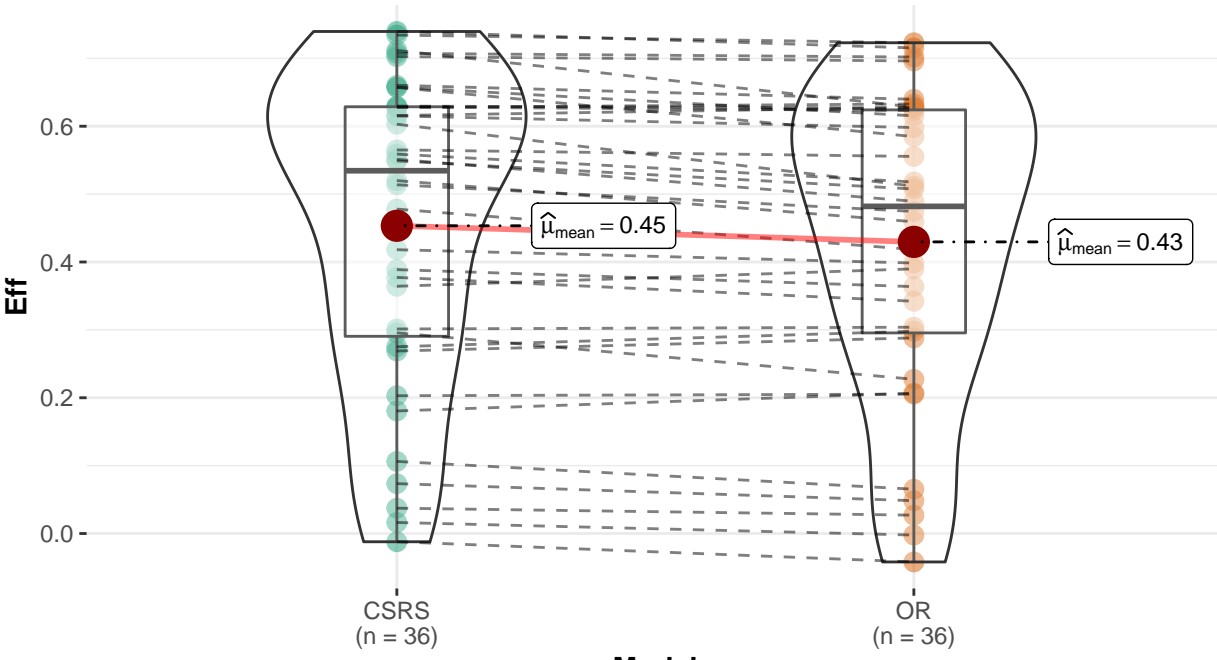

$\log_e(\text{BF}_{01}) = -6.22, \widehat{\delta}_{\text{difference}}^{\text{posterior}} = 0.02, \text{CI}_{95\%}^{\text{HDI}} [0.01, 0.03], r_{\text{Cauchy}}^{\text{JZS}} = 0.71$

## CBS CSRS vs OR: Longitudinal

$t_{\text{Student}}(35) = 11.44, p = 2.25\text{e}{-}13, \widehat{g}_{\text{Hedges}} = 1.87, \text{CI}_{95\%} [1.34, 2.43], n_{\text{pairs}} = 36$

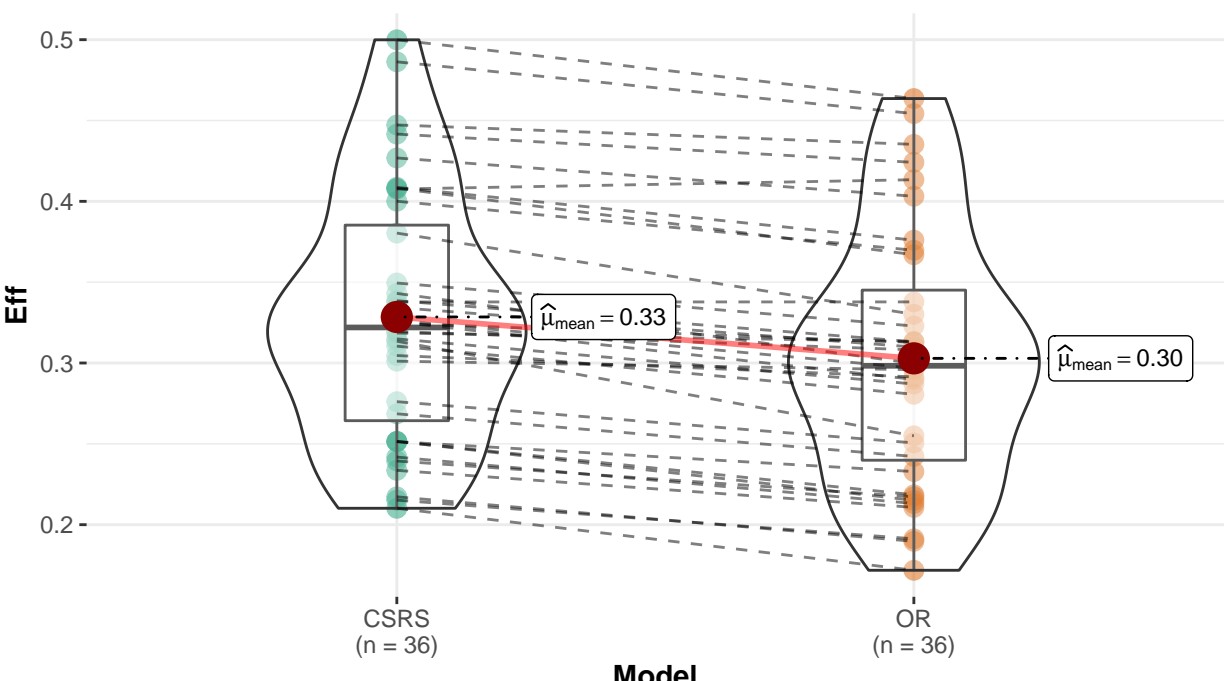

$\log_e(\text{BF}_{01}) = -24.25, \widehat{\delta}_{\text{difference}}^{\text{posterior}} = 0.03, \text{CI}_{95\%}^{\text{HDI}} [0.02, 0.03], r_{\text{Cauchy}}^{\text{JZS}} = 0.71$

### 5.3.5 Top $n$ effect sizes and pairwise tests for PSP

These regions are largely consistent with a priori hypotheses/ PSP overall may require further investigation in terms of understanding disease effects and how these are reflected within standard processing approaches used here.

Table 33: OR PSP : regions sorted by cross-sectional effect sizes.

|  | Anat | DXPSP | deltaTime.DXPSP |
|---|---|---|---|
| 421 | vol_bn_gp_gpi_leftcit168 | 1.0736670 | 0.0934710 |
| 424 | vol_bn_gp_gpi_rightcit168 | 1.0360773 | -0.0398132 |
| 415 | vol_bn_gp_gpe_leftcit168 | 0.6081861 | 0.1845209 |
| 109 | vol_left_precentraldktcortex | 0.6040893 | 0.1225103 |

Table 34: CSRS PSP : regions sorted by cross-sectional effect sizes.

|  | Anat | DXPSP | deltaTime.DXPSP |
|---|---|---|---|
| 423 | vol_bn_gp_gpi_leftcit168 | 1.1541795 | 0.1778124 |
| 426 | vol_bn_gp_gpi_rightcit168 | 1.1285760 | -0.0470690 |
| 417 | vol_bn_gp_gpe_leftcit168 | 0.7530286 | 0.1826525 |
| 420 | vol_bn_gp_gpe_rightcit168 | 0.7269553 | 0.1320500 |

Table 35: OR PSP : regions sorted by longitudinal effect sizes.

|  | Anat | DXPSP | deltaTime.DXPSP |
|---|---|---|---|
| 151 | vol_right_inferior_parietaldktcortex | 0.2797533 | 0.3300449 |
| 130 | vol_left_supramarginaldktcortex | 0.4687684 | 0.2984942 |
| 58 | vol_left_inferior_parietaldktcortex | 0.1548217 | 0.2721653 |
| 217 | vol_right_superior_parietaldktcortex | 0.2767687 | 0.2670540 |

Table 36: CSRS PSP : regions sorted by longitudinal effect sizes.

|  | Anat | DXPSP | deltaTime.DXPSP |
|---|---|---|---|
| 132 | vol_left_supramarginaldktcortex | 0.4817732 | 0.3287657 |
| 153 | vol_right_inferior_parietaldktcortex | 0.3002098 | 0.3111182 |
| 60 | vol_left_inferior_parietaldktcortex | 0.1756202 | 0.2892939 |
| 219 | vol_right_superior_parietaldktcortex | 0.3178136 | 0.2775486 |

Table 37: PSP : cross-sectional CSRS vs OR pairwise t-tests (continued below)

| Test statistic | df | P value | Alternative hypothesis |
|---|---|---|---|
| 5.505 | 35 | 3.472e-06 * * * | two.sided |

| mean of the differences |
| --- |
| 0.03848 |

Table 39: PSP : longitudinal CSRS vs OR pairwise t-tests (continued below)

| Test statistic | df | P value | Alternative hypothesis |
| --- | --- | --- | --- |
| 4.204 | 35 | 0.0001722 * * * | two.sided |

| mean of the differences |
| --- |
| 0.01407 |

### PSP CSRS vs OR: cross−sectional

$t_{\text{Student}}(35) = 5.50$, $p = 3.47\text{e}{-}06$, $\widehat{g}_{\text{Hedges}} = 0.90$, $\text{CI}_{95\%}$ [0.52, 1.29], $n_{\text{pairs}} = 36$

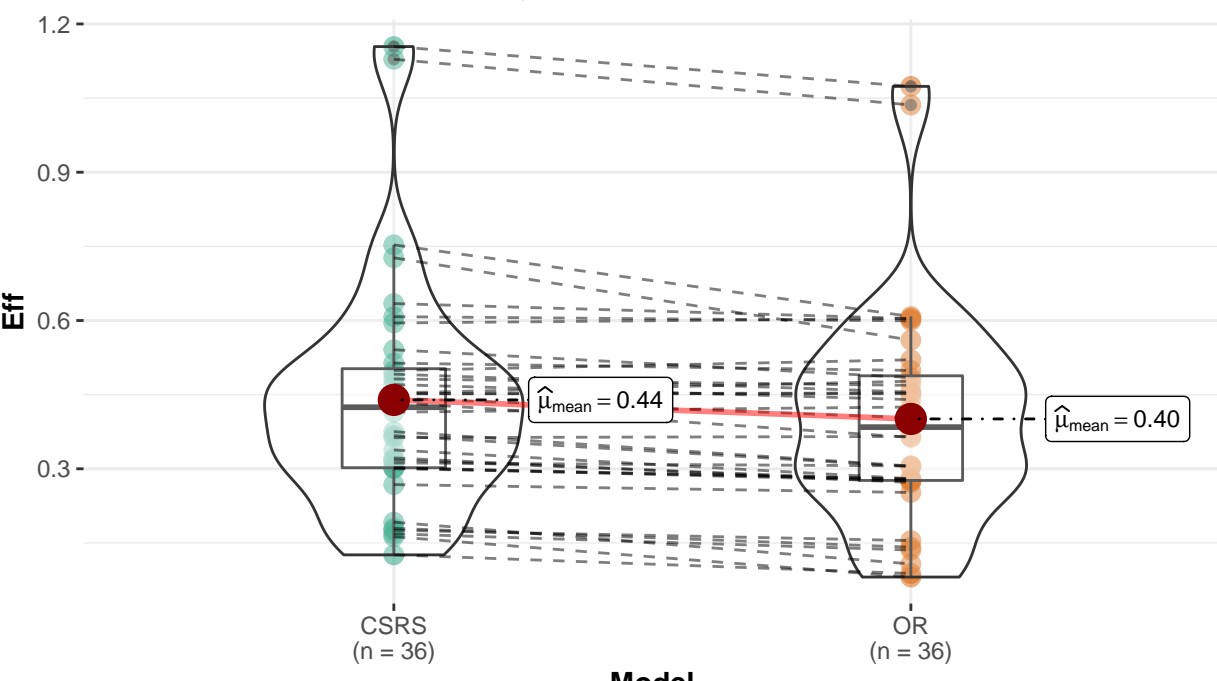

$\log_e(\text{BF}_{01}) = -8.60$, $\widehat{\delta}_{\text{difference}}^{\text{posterior}} = 0.04$, $\text{CI}_{95\%}^{\text{HDI}}$ [0.02, 0.05], $r_{\text{Cauchy}}^{\text{JZS}} = 0.71$

# PSP CSRS vs OR: Longitudinal

$t_{\text{Student}}(35) = 4.20$, $p = 1.72\text{e}{-}04$, $\widehat{g}_{\text{Hedges}} = 0.69$, $\text{CI}_{95\%}$ [0.33, 1.05], $n_{\text{pairs}} = 36$

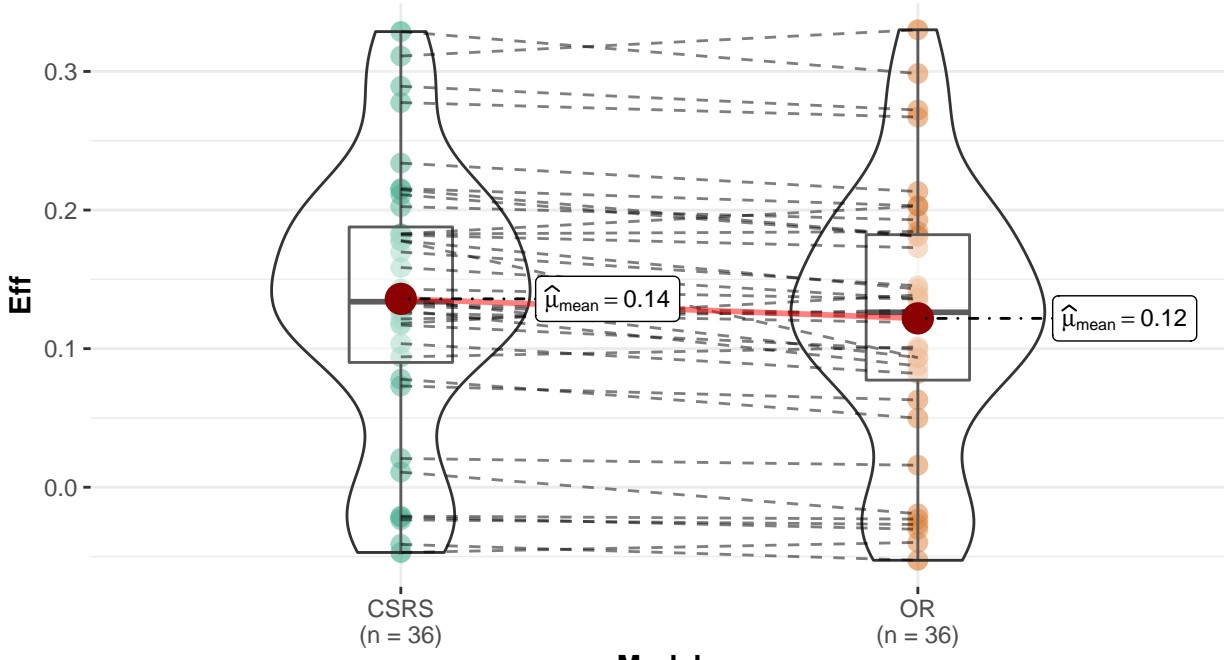

$\log_e(\text{BF}_{01}) = -5.03$, $\widehat{\delta}_{\text{difference}}^{\text{posterior}} = 0.01$, $\text{CI}_{95\%}^{\text{HDI}}$ [6.06e−03, 0.02], $r_{\text{Cauchy}}^{\text{JZS}} = 0.71$

## 5.4 other performance implications with respect to individual regions and diagnoses

We also examined relationships between the volume or thickness of the input region and the effect size and effect size improvement. Overall, smaller regions tend to show greater improvement, as expected. However, this effect is limited to regions below a given threshold of size. As we do not have a continuous distribution of region size in these data, more work would be needed to identify both size and shape criterion to help determine which types of regions should improve most under CSRS or related SR/PSR methods. These results may vary with the disease and, as such, could be considered in a disease-specific and region-specific manner. It is likely that one could identify results that are superior to those identified here if one used targeted, goal-specific training instead of the disease and region agnostic strategy that we employed.

# References

1. Annonny Z. Supplemental information : Concurrent 3D super resolution on intensity and segmentation maps improves detection of structural effects in neurodegenerative disease. 2022.