# OpenReview forum: "Concurrent 3D super resolution on intensity and segmentation maps improves detection of structural effects in neurodegenerative disease"
_NeurIPS.cc/2022/Conference — NeurIPS 2022 Submitted_

### Official Review · Reviewer_Uzm9 · 2022-06-23

**Rating:** 4
**Confidence:** 3
**Soundness:** 2 fair
**Presentation:** 2 fair
**Contribution:** 1 poor

**Summary:**

This paper proposed a 2x (from 2mm to 1mm) super-resolution method for 3D medical image volumes and the corresponding segmentation mask for better brain diagnosis. The proposed method is a natural extension of the 2D super-resolution model called deep back-projection network (DBPN). Compared with linear upsampling and nearest-neighbor upsampling, the proposed method achieved a better dice score as well as a better visual quality.

**Questions:**

Is this the first super-resolution 3D model in this area? If not, can you make a comparison between the proposed method and the existing methods?

**Limitations:**

As stated in the weakness, the method is not novel to me and the experimental comparison is not satisfactory for a NeuroIPS paper. The paper lacks a thorough description of related work and a better comparison with other existing methods other than linear/nearest-neighbor upsampling.

**Strengths And Weaknesses:**

Strength:
This paper provides good implementation/technical details to illustrate the problem, the method, and the experiments. It also provides a good visual-based comparison to interpret the method's performance. It is interesting to consider upsampling both the image along with the segmentation mask.


Weakness:
The method proposed in the paper doesn't seem to be novel given the fact that it is a natural extension from 2D to 3D based on the deep back-projection network (DBPN). Though the paper proposed a few loss functions to adjust the model's objective, the performance does not show a significant difference between different combinations of loss functions.

Moreover, the paper doesn't provide a good description of the related work and the main comparison is done between the variants of the proposed method and the trivial methods (linear/nearest-neighbor upsampling).

---

> ### Author Response · Authors · 2022-08-02
> **brief responses**
>
> re: *the performance does not show a significant difference between different combinations of loss functions*.
>
> We disagree here.  See the confidence intervals (boot.95ci) in Table 1.  E.g. the best model’s CI’s do not overlap with several other models.  And it outperforms baseline (original resolution) from 5 to 20 percent depending on the region.
>
>
> **Is this the first super-resolution 3D model in this area? If not, can you make a comparison between the proposed method and the existing methods?**
>
> It is not the first PSR model, as noted in the text.  However, it is the first evaluation that connects to clinical methods and likely the first demonstration of (1) novel perceptual loss functions that are intrinsically 3D and (2) experiments that show these losses are practically useful and as good or better than other alternatives.
>
> We did attempt to use SOUP-GAN but found very little support for the method and that its associated package will not load basic required functionality in python.   We also sought control over the many possible parameters that could lead to uninterpretable differences in performance.

---

### Official Review · Reviewer_kpkJ · 2022-07-11

**Rating:** 7
**Confidence:** 4
**Soundness:** 3 good
**Presentation:** 4 excellent
**Contribution:** 2 fair

**Summary:**

The authors present an extension of a 2D perceptual super-resolution architecture extended to 3D within neuroimaging. The authors additionally extend the network to simultaneously produce up-sampled segmentation maps. The upsampled segmentation maps are evaluated against the ground truth, as well as against detection power across a variety of frontotemporal disorders. The effect of different loss functions used in training are also evaluated against prediction power. Whilst the super-resolution techniques improved standard evaluation metrics (PSNR & SSIM) when compared against the low resolution data, these metrics did not significantly improve when compared against models trained with the different loss functions used. Prediction power for frontotemporal disorders was improved in many regions when using the super-resolution models, however some regions suffered in prediction power when super-resolved.



**Questions:**

I have no questions at this point.

**Limitations:**

The authors have addressed the limitations of their work in the conclusions section.

**Strengths And Weaknesses:**

Overall this paper is clear and concise, and forms a well constructed manuscript. The background information and methods are extensive, and are presented in logical, clearly defined, sections. The overall study is fairly in depth for the given research question, and presents a nice analysis of the downstream effect of segmentation super-resolution within a disease model. The authors include the relevant previous work done in this area, and appropriately outline their contribution to this field.

The biggest limitation of this work I feel is the limitation of impact in some of novel contributions outlined. Specifically, extending a model from 2D to 3D, whilst relevant within neuroimaging data, is not particularly impactful. Additionally, adding segmentation maps as an additional output channel does improve user convenience, and likely a computational efficiency gain, but is ultimately not overly impactful as the segmentation maps could be re-calculated with the inferred high-resolution dataset. Overall however, it is my opinion that the strengths do clearly outweigh the weaknesses in this work.

---

### Official Review · Reviewer_hYgi · 2022-07-16

**Rating:** 3
**Confidence:** 4
**Soundness:** 2 fair
**Presentation:** 1 poor
**Contribution:** 1 poor

**Summary:**

This work developed a super resolution method for 3D neuroimaging and evaluate its performance in detecting brain changes due to neurodegenerative disease. It trained on 3D brain data to upsample both the raw intensity image and associated segmentation labels. The method was mainly based on the 2D deep back projection network (DBPN) [5], which was extended to three dimension and multiple output. The method was evaluated in downstream clinically relevant signal detection problem: quantifying cross-sectional and longitudinal
change across a set of phenotypically heterogeneous but related disorders that exhibit known and differentiable patterns of brain atrophy.

**Questions:**

See the detailed comments above.

**Ethics Review Area:**

["I don’t know"]

**Limitations:**

No.

**Strengths And Weaknesses:**

Strength:
-- This work conducted the evaluation of super resolution in the downstream clinically relevant tasks, e.g., quantifying cross-sectional and longitudinal change across a set of phenotypically heterogeneous but related disorders.

Weakness:
-- This work overall has limited methodology contribution.

-- The organization of this work is not clear and hard to follow for some sections.

-- This work fits more to a dedicated medical imaging conference or journal.

---

### Official Review · Reviewer_Ho1b · 2022-07-19

**Rating:** 3
**Confidence:** 4
**Soundness:** 2 fair
**Presentation:** 1 poor
**Contribution:** 1 poor

**Summary:**

The paper presents a new methodology for concurrent super resolution and segmentation (CSRS) on 3-D volumetric MRI data to consistently upsample both an image intensity channel and associated segmentation labels. To this end, a primary contribution is in adapting perceptual super resolution frameworks designed for 2-D data to 3-D data within the 2D deep back projection (DBPN) network.

Since the ground truth high resolution images may not be present for a given dataset, the authors propose an indirect evaluation via the quantification of cross-sectional and longitudinal change in diseased cohorts. Specifically, they choose a set of phenotypically heterogeneous but related disorders that are associated with known patterns of brain atrophy. Their experiments examine the effect of various choices in loss function in terms of identification of neurodegenerative diseases within the Human Connectome Project (HCP), Data: Parkinson’s Progression Markers Initiative (PPMI) , Frontotemporal Lobar Degeneration Neuroimaging Initiative (NIFD) & and-Repeat
123 Tauopathy Neuroimaging Initiative (4RTNI) datasets.

**Questions:**

1. Section 2.1 : "We then set weights relative to the value of the reconstruction error after convergence such that: the TV loss is roughly 2/3 the reconstruction term (R); the perceptual loss is roughly 3x R; the Dice loss is roughly equivalent to the perceptual loss. This strategy, based on our task-specific goals, enables us  to compare models consistently and add/subtract terms without extensive weight optimization."

How did the authors arrive at this heuristic?

2. Formatting issues

(a) Oddly large spacing after section 2 heading

(b) The plots in the paper are a bit blurry and tables are included within figures. A suggestion would be to replace these with better resolution images and use the default latex package for displaying the tabular data.

3. Discussion "we trained each model for 2 epochs or until convergence", does this imply that the maximum number of epochs used is 2?

**Limitations:**

The main limitations discussed in section 6 including computation and quality of annotations. Given that the framework is so computationally expensive that 1 epoch takes 12 hours seems to greatly limit the practical utility.

**Strengths And Weaknesses:**

STRENGTHS: The clinical problem that the paper examines, concurrent super resolution and segmentation is interesting, as is the use-case  of tracking brain atrophy within neuro-degenetative diseases. The proposed approach to extending existing Perceptual Super Resolution techniques is interesting, though a bit incremental on the technical front.

WEAKNESSES:

1. The evaluation section is rather poorly explained and hard to follow. Several points are unclear and the main arguments are not very convincing in light of the quantitative results
(a) Some key details such as dataset splits for training, testing, validation are not clearly mentioned.
(b) It is unclear which dataset/disorder (all datasets?) the results in Table 1 and 2 and figures correspond to.
(c) The differences across different loss functions and configurations in Table 1 are rather minor (third decimal place in terms of effect sizes, dice, psnr). It is unclear whether these improvements are consistent across dataset splits.
(d)  Table 2 is really hard to parse, lacking in description and in general confusing. As per my understanding, CSRS.R.TV.VGG vs the proposed framework should be compared across the same anatomical class. If so, the differences in effect sizes appear very minor (third decimal place) for several comparisons.

2. If my understanding is correct, the baseline comparisons in the paper correspond to evaluation against ablated versions of the framework (various loss functions), evaluation in the original resolution and linear interpolation. Given that the authors listed a couple of recent approaches proposed in literature in Section 1, it is not obvious why they did not include these as baseline comparisons.

---

> ### Author Response · Authors · 2022-08-02
> **brief responses**
>
> the reviewer appears to have some confusion about training/testing and validation data.   the methods were trained on HCP data.   the methods were then applied to independent data in frontotemporal dementia ... in these data, we perform statistical inference in the style of "ecological" or "real world" evaluation that enables us to assess detection power.
>
> summary statistics in Table1 aggregate over all data.  Table 2 annotates which diseases are associated with the listed result.  More detail is in supplementary information.
>
> re: *The differences across different loss functions and configurations in Table 1 are rather minor (third decimal place in terms of effect sizes, dice, psnr).*
>
> This is an incorrect observation.  Please look at the results for PSNR, primary effect size results and t-statistics.  Importantly, look closely at Figure 5 which shows very good detection performance improvement for many regions.
>
>
> **How did the authors arrive at this heuristic?**
>
> This is based on reference loss weights from the original DBPN papers.
>
> **does this imply that the maximum number of epochs used is 2?**
>
> A minimum of 2 epochs.  We note that little improvement can occur in less than 1 epoch for some models due to the large number of patch samples that we extracted.  I.e. simpler (fewer loss term) models converge more quickly and the patch samples (though not repeated) contain substantial redundant information.  So the notion of "epoch" is not particularly meaningful here.  I.e. one may reach effective convergence without seeing every sample.

---

> > ### Comment · Reviewer_Ho1b · 2022-08-06
> > **Response to Author Reply**
> >
> > Thank you for your reply. Some points have been partially clarified, for example, training vs validation/testing setup. However, I am not sure I entirely agree with the response that the differences in Table 1 are not marginal. Taking the SSIM or dice.mean columns as an example, I still see that the entire suite of comparison methods (various configurations of the CSRS model) are very close.
> >
> > Nevertheless, I still think that the content of the paper needs major restructuring and clearer presentation. I also agree with comments made by other reviewers regarding the incremental methodological contribution.

---

### Official Review · Reviewer_aKcw · 2022-07-20

**Rating:** 3
**Confidence:** 4
**Soundness:** 2 fair
**Presentation:** 2 fair
**Contribution:** 2 fair

**Summary:**

This paper outlines a new method for enhancing 3D neuroimaging resolution and improving image segmentation using "perception super resolution" in a technique labelled "concurrent super resolution and segmentation" (CSRS). The authors evaluate the effectiveness of the technique on publicly available clinical datasets from the Human Connectome Project, the Parkinson's Progression Markers Initiative, and the Frontotemporal Lobar Degeneration Neuroimaging Initiative (NIFD) & 4-Repeat Tauopathy Neuroimaging Initiative (4RTNI). They test this method in quantifying changes across cross-sectional and longitudinal MRI datasets for different neurodegenerative diseases. The results, however, only report comparisons of the new technique to the original-resolution model using bootstrapped t-tests, which are not corrected for multiple testing. The implications are very briefly discussed.

**Questions:**

I do not have any questions

**Limitations:**

The authors do not address the limitations or potential negative social impact of their work.

**Strengths And Weaknesses:**

I do not comment on paper originality as it is a flawed measure of paper quality.
The overall quality of the paper is moderate; while the technique is interesting, there is insufficient detail about preprocessing of the MRI data to replicate the results, and the methods are not communicated very clearly. In addition, more of a clinical and/or neuroscience perspective in the experimental design would have been useful. The Discussion is overly brief and lacking in depth or detail. The Supplementary Materials are quite poorly organized and do not meet NeurIPS communication standards. However, the technique itself does seem very promising, and with clearer communication, this could be an interesting and important development.

---

> ### Author Response · Authors · 2022-08-02
> **Re: there is insufficient detail about preprocessing of the MRI data**
>
> The baseline MRI analysis is described in reference 16 Tustison et al 2021 -- this refers to a broadly used toolkit that is fairly standard in medical image processing.

---

> > ### Comment · Reviewer_aKcw · 2022-08-07
> > **This should have been more clearly flagged in the paper**
> >
> > If this is the case, then the authors should have made this clearer in the paper; for instance, "for full details of preprocessing pipeline, see [16]" or similar.

---

### Official Review · Reviewer_CkcC · 2022-07-21

**Rating:** 6
**Confidence:** 3
**Soundness:** 3 good
**Presentation:** 3 good
**Contribution:** 3 good

**Summary:**

This work introduces a new perceptual super resolution method for 3D brain image segmentation. The method uses a carefully designed up-sampling and a novel loss function to obtain the desired performance, and this method is evaluated based on a clinical relevant metric due to the lack of high resolution ground truth data. The authors find that the proposed method consistently improves the ability to detect regional atrophy both longitudinally and cross-sectionally in five relevant diseases.

**Questions:**

Please refer to the "weakness" part.

**Limitations:**

Yes.

**Strengths And Weaknesses:**

Strength:
- This work addresses an important problem in the field of medical imaging, and the authors have demonstrated deep understanding and expertise in this field.
- The method is discussed in a thorough manner, and important decisions are well justified. Truly quality work!
- Rigorous experiments to evaluate the proposed method's performance (it's great to see CI evaluations)

Weakness:
- This work would be of a stronger stance if its originality is more well-justified: 1) the paper writes that "...three perceptual loss functions for PSR, two of which are new", while the neural network and the loss functions used (reconstruction error, TV, perceptual loss, dice) are not unheard in the field of medical imaging. The authors might elaborate which parts of the loss functions are new. 2) While the proposed evaluation paradigm is, in a sense, new, I would like to see a deeper discussion of why this is new (e.g., is this a novel way of seeing the problem of lacking SR ground truth label?)
- The contribution bullet states that "demonstration of the impact of loss choice on performance differences in improving detection power in [population studies] of neurodegenerative disease". The presented results haven't directly shown the power of the proposed method on a population-scale. The authors might elaborate more about the stated population-level effect of the proposed method.

---

> ### Author Response · Authors · 2022-08-02
> **brief responses to questions**
>
> * Yes, this is a clinically-driven way of assessing the impact of PSR on medical image processing.  We are not aware of any published work that has performed such an extensive and controlled study that links PSR with potential clinical trial impact.
>
> * population analysis - We apologize for the confusion with respect to the use of this terminology.  We intend “population study” to mean a statistical analysis of a population sample for these diseases.

---

> > ### Comment · Reviewer_CkcC · 2022-08-04
> > **reply to authors**
> >
> > - I wonder if this study is truly a controlled trial study? In this case there would be a carefully recruited controlled group, whose only difference with the test groups is not being exposed to the substance to be tested, ideally. Simply using positive/negative labels in a sample usually is not considered as a controlled study.
> >
> > - Thanks for the clarification. I recommend the authors to change the wording, as "population analysis" might sound overclaiming in this case.

---

### Author Response · Authors · 2022-08-02
**Overall response to reviewer comments**

**General response/context to submission**: We feel strongly that the medical research field and the broader Neurips community would benefit from more cross-pollination.  Discussions similar to those that this submission has sparked with reviewers would increase synergy between these related communities. *Improving the relevance of Neurips to broader society (including medicine) is an important topic* and promoting cross-domain research is one way to achieve this.  This was one goal of the submission: to raise awareness in Neurips of why and how one might apply SR to medical images.

**Evaluating PSR with respect to clinical utility for disease trials**: Our primary evaluation outcome is improvement in neurodegeneration related cross-sectional and longitudinal effect sizes due to PSR.  Critically, brain structure changes (captured here by segmentation) are one of the key signatures of the frontotemporal disorders and play a key role in assessing the impact of potential interventions.
* The improvement that we show can translate, in theory, to better designed and more powerful clinical trials which can reduce patient and societal burden.  In some regions/disorders, effect sizes improve by 20% or more.  This would lead to a substantial increase in power for a fixed sample size if these findings can be translated to clinical studies.
* The disorders we study currently have no treatment, though several are under development.  As such, efficient trials are important to pursue.  And our study focuses on regions that would be specifically relevant for each type of frontotemporal disorder under consideration.
* We include SSIM and PSNR as these are standard in SR papers.  However, we note that these outcomes do not vary much in comparison to detection power.  This demonstrates an additional limitation to PSNR and SSIM in the context of SR for medical image quantification.  I.e. they may not reflect impact on image quantification.

**Primary technical contribution**: To achieve the goal of assessing clinical utility of PSR, we compared — with fixed underlying super-resolution architecture — several different loss formulations. The best perceptual features in this application domain arise from a (shallow layer of a) ResNet trained to predict human ratings of image quality.  *Reliable perceptual features in 3D are important to document, compare and disseminate due to their broad potential utility* within and beyond PSR.   We note that our best model uses this new perceptual feature layer and its detection performance differences with other PSR models are substantial as evidenced by Table 1 column boot.95ci.  The bootstrapping estimated upper limits of other well-performing models (e.g. CSRS.R.TV.VGG) are near the lower limit of the best model (CSRS.R.TV.D.Res6).

**Reference results**:  Reference implementations used in the paper include:
 * the baseline results - this is the reference against which everything else is compared and would be considered the standard analysis procedure;  the baseline analysis is described in reference 16 Tustison et al 2021.
 * the VGG19 pseudo-3D perceptual loss used in SOUP-GAN (reference 12 in original paper)
 * The simplest SR intensity loss: only the mean squared intensity difference with a small denoising component from total variation (constant in all experiments).
 * Basic linear upsampling: necessary for this foundational experiment as it is the most commonly used in practice.
 We kept architecture constant to isolate the effect of the perceptual loss.


**Closing remarks**: We thank the reviewers for their efforts and we understand their concerns which are overall reasonable.  We hope these additional summary notes help sharpen the context and contributions of this submission for reviewers.  Detailed responses will be inline for each review, as needed.

---

### Author Response · Authors · 2022-08-08
**discussion period changes to submitted pdf**

we revised our contribution bullet points to read:

demonstration that loss choice impacts detection power in \textcolor{blue}{natural history studies of neurodegenerative disease.  Standard intensity similarity and segmentation overlap metrics, on the other hand, do not discriminate performance between the candidate CSRS options.}

and explicitly added a statement about MRI processing details:

All MRI processing details may be found in [16].


We submitted a revised pdf where these changes are shown in blue.

---

### Meta-Review · Area_Chair_StaN · 2022-08-24

**Recommendation:** Reject
**Confidence:** Certain

**Metareview:**

This paper has mixed evaluations, with two reviewers recommending accept and three recommending reject. After carefully reading the paper and the discussion, I agree with reviewers hYgi, Ho1b, aKcw, Uzm9 in their main criticisms. The paper still requires major revisions before it can be accepted, including, but not limited to, an improvement in the clarity of the presentation and more experimental comparisons against other, perhaps, even simpler approaches.

**Award:**

No

---

### Decision · Program_Chairs · 2022-09-14

Reject